# Sparse Spectral Training and Inference on Euclidean and Hyperbolic Neural Networks

## Abstract

The growing demands on GPU memory posed by the increasing number of neural network parameters call for training approaches that are more memory-efficient. Previous memory reduction training techniques, such as Low-Rank Adaptation (LoRA) and ReLoRA, face challenges, with LoRA being constrained by its low-rank structure, particularly during intensive tasks like pre-training, and ReLoRA suffering from saddle point issues. In this paper, we propose **Sparse Spectral Training (SST)** to optimize memory usage for **pre-training**. SST *updates all singular values* and *selectively updates singular vectors* through a multinomial sampling method weighted by the magnitude of the singular values. Furthermore, SST employs *singular value decomposition to initialize and periodically reinitialize* low-rank parameters, reducing distortion relative to full-rank training compared to other low-rank methods. Through comprehensive testing on both Euclidean and hyperbolic neural networks across various tasks, including natural language generation, machine translation, node classification, link prediction, and image classification, SST demonstrates its ability to outperform existing memory reduction training methods and is comparable to full-rank training in various cases. On LLaMA-1.3B, with only 18.7% of the parameters trainable compared to full-rank training (using a rank equivalent to 6% of the embedding dimension), SST reduces the perplexity gap between other low-rank methods and full-rank training by **97.4%**. This result highlights SST as an effective parameter-efficient technique for model pre-training, offering a promising new paradigm for achieving scalable and memory-efficient neural network training. Our code is available at `https://anonymous.4open.science/r/sparse_spectral_training-6A2C/`.

## 1 Introduction

The development and scaling up of large language models (Kaplan et al., 2020; Brown et al., 2020; Touvron et al., 2023b) pose great challenges to the feasibility of training large language models from scratch. Normal training methods that update all parameters of models become extremely expensive due to their extensive GPU memory requirements.

Recent developments in parameter-efficient fine-tuning (PEFT) methods, such as Low-Rank Adaptation (LoRA) (Hu et al., 2022), have sought to mitigate the challenge of fine-tuning memory requirements by introducing trainable low-rank matrices that efficiently reduced the memory footprint. However, limiting the model's parameter updates to a *low-rank subspace* can severely restrict the ability of a model to capture and represent complex data patterns, leading to suboptimal performance, especially in the pre-training stages. Recent advancements such as ReLoRA (Lialin et al., 2024), COLA (Xia et al., 2024), and PLoRA (Meng et al., 2024b) have addressed the limitation of low-rank constraint, by iteratively merging low-rank parameters with frozen parameters. However, they still encounter *saddle point issues* due to zero gradient of low-rank parameters that occurs after each merging step. This challenge results in slower and less effective convergence compared to full-rank models during pre-training.

In response to these challenges, we introduce **Sparse Spectral Training (SST)**, a new training framework designed to optimize memory consumption while closely approximating the overall learning dynamics and performance of full-rank training. Unlike previous methods (Hu et al., 2022; Lialin et al., 2024; Zhang et al., 2023; Ding et al., 2023) that primarily focus on updating within

a low-rank subspace at each step, SST adopts a more effective approach by **updating all singular values at each step**. SST also leverages the intrinsic spectral properties of the weight matrices, focusing **selective updates of singular vectors** sampled from a multinomial distribution weighted by the magnitude of the singular values. Additionally, SST uses **singular value decomposition to initialize and reinitialize** low-rank parameters during training, reducing distortion relative to full-rank training compared to other low-rank methods.

Our comprehensive evaluations cover different tasks, including pre-training large language models on OPT model family, ranging from 125m to 1.3b (Zhang et al., 2022), using Transformer (Vaswani et al., 2017) for machine translation tasks and hyperbolic graph neural networks (Chen et al., 2022) on node classification and link prediction tasks. For the OPT and LLaMA model family, SST reduces the perplexity gap between other low-rank methods and full-rank training by **50%-97.4%**. In machine translation with Transformers, SST reduces the BLEU gap by an average of **66.7%**. Furthermore, we are the first to incorporate parameter-efficient pre-training process in hyperbolic space, demonstrating that SST is a general technique applicable across various data structures and models. On the hyperbolic Transformer, SST even **outperforms full-rank training** in most scenarios. For hyperbolic graph neural networks, SST reduces the performance gap by an average of **73.7%** in node classification and **82.5%** in link prediction.

## 2 RELATED WORK

**Low-Rank Adaptation.** Low-rank adaptation has become a key strategy for reducing the computational and memory requirements of training large-scale neural networks. Hu et al. (2022) introduced Low-Rank Adaptation (LoRA), a technique that fine-tunes pre-trained models by integrating low-rank matrices to significantly reduce the number of parameters updated during training. Various enhancements to LoRA have since been developed to improve its efficiency and broaden its application (Zhang et al., 2023; Dettmers et al., 2024; Zi et al., 2023; Valipour et al., 2023). Lialin et al. (2024) introduced ReLoRA specifically for the pre-training phase, which requires a full-rank warm-up to achieve performance comparable to full-rank training. A similar approach is also found in COLA (Xia et al., 2024) and PeriodicLoRA (Meng et al., 2024b). Additionally, Zhao et al. (2024) introduced GaLore, which projects gradients into a low-rank subspace. Meng et al. (2024a) introduced PiSSA, which applies SVD-based low-rank updates for fine-tuning pre-trained weights by focusing on dominant singular vectors. These advancements highlight the versatility and ongoing evolution of low-rank adaptation techniques in response to the growing complexity of neural network models. Other parameter-efficient training methods are included in Appendix C.

**Hyperbolic Neural Networks.** Hyperbolic neural networks are an emerging area in deep learning, exploiting the unique properties of hyperbolic space that make it ideal for processing hierarchical and graph-structured data (Muscoloni et al., 2017; Cannistraci & Muscoloni, 2022). Innovations in this area have adapted fundamental neural network mechanisms to function within hyperbolic geometries, as demonstrated by Muscoloni et al. (2017) and Ganea et al. (2018). Further developments by Chen et al. (2022) explore manifold-specific properties to enrich both theoretical understanding and practical deployment. The use of hyperbolic spaces has been shown to significantly improve data representation and generalization across various tasks, marking a notable advancement in managing complex, non-Euclidean data structures (Gulcehre et al., 2019; Liu et al., 2019; Tifrea et al., 2019).

## 3 LOW RANK ADAPTATION

This section introduces the fundamentals and limitations of Low-Rank Adaptation (LoRA) (Hu et al., 2022), ReLoRA (Lialin et al., 2024), and GaLore (Zhao et al., 2024). These limitations are addressed by Sparse Spectral Training (SST) in Section 4.

### 3.1 LoRA

LoRA (Hu et al., 2022) fine-tunes a pre-trained model by learning an incremental update $\Delta\mathbf{W}$ to the pre-trained and frozen pre-trained weight matrix $\mathbf{W}_0$. Here, $\mathbf{W}_0, \Delta\mathbf{W} \in \mathbb{R}^{m \times n}$ with $m \leq n$. LoRA decomposes $\Delta\mathbf{W}$ into the product of two low-rank matrices, $\mathbf{B} \in \mathbb{R}^{m \times r}$ and $\mathbf{A} \in \mathbb{R}^{r \times n}$, such that $\Delta\mathbf{W} = \mathbf{B}\mathbf{A}$. This decomposition is applied to a linear layer with input $\mathbf{x}$ and output $\mathbf{h}$ as follows:

$$\mathbf{h} = (\mathbf{W}_0 + \Delta\mathbf{W})\mathbf{x} = (\mathbf{W}_0 + \mathbf{BA})\mathbf{x} \tag{1}$$

Given $r \ll min(m, n)$, LoRA significantly reduces GPU memory usage compared to full-rank fine-tuning.

**Limitation of LoRA.** Consider $\mathbf{W}^*$ as the optimal weight matrix which minimizes loss. The deviation from the current weights is $\Delta\mathbf{W}^* = \mathbf{W}^* - \mathbf{W}_0$. Performing a singular value decomposition on $\Delta\mathbf{W}^*$ yields $\Delta\mathbf{W}^* = \mathbf{U}\boldsymbol{\Sigma}\mathbf{V}^{\mathrm{T}}$, where $\mathbf{U} \in \mathbb{R}^{m \times m}$, $\boldsymbol{\Sigma} \in \mathbb{R}^{m \times m}$, $\mathbf{V}^{\mathrm{T}} \in \mathbb{R}^{m \times n}$.

$\mathbf{U}$ and $\mathbf{V}^{\mathrm{T}}$ are orthonormal bases, $\mathbf{U} = [\mathbf{u}_1, \mathbf{u}_2, ..., \mathbf{u}_m]$, $\mathbf{V} = [\mathbf{v}_1, \mathbf{v}_2, ..., \mathbf{v}_m]$. $\boldsymbol{\Sigma}$ is a diagonal matrix with entries $\{\sigma_1, \sigma_2, ..., \sigma_m\}$. Then the Eckart–Young–Mirsky theorem (Eckart & Young, 1936) states:

$$\|\Delta\mathbf{W}^* - \Delta\mathbf{W}\|_{\mathrm{F}} \geq \sqrt{\sigma_{r+1}^2 + \cdots + \sigma_m^2} \tag{2}$$

where $\|\mathbf{W}\|_{\mathrm{F}} = \sqrt{\sum_{i=1}^{m}\sum_{j=1}^{n} w_{ij}^2}$ is the Frobenius norm, with $w_{ij}$ being the element at row $i$ and column $j$ of $\mathbf{W}$. Equality holds when $\mathbf{B} = [\sqrt{\sigma_1}\mathbf{u}_1, \sqrt{\sigma_2}\mathbf{u}_2, ..., \sqrt{\sigma_r}\mathbf{u}_r]$ and $\mathbf{A}^{\mathrm{T}} = [\sqrt{\sigma_1}\mathbf{v}_1, \sqrt{\sigma_2}\mathbf{v}_2, ..., \sqrt{\sigma_r}\mathbf{v}_r]$. This suggests that LoRA can only closely approximate the performance of full-rank training in simple tasks like fine-tuning, where $\sigma_i \approx 0, i \in \{r+1, ..., m\}$. However, in more complex scenarios like pre-training, where $\sigma_i, i \in \{r+1, ..., m\}$ are non-negligible, LoRA may struggle to achieve the same level of performance as full-rank training.

### 3.2 ReLoRA*

ReLoRA (Lialin et al., 2024), COLA (Xia et al., 2024), and PLoRA (Meng et al., 2024b) address the limitation of fixed low ranks by iteratively merging the low-rank matrices $\mathbf{B}$ and $\mathbf{A}$ back into the base weight matrix $\mathbf{W}_0$. Although ReLoRA is designed for pre-training, it includes an initial period of full-rank training (referred to as a "warm start"), which prevents it from being fully end-to-end parameter-efficient. Meanwhile, COLA and PLoRA are primarily intended for fine-tuning. In this paper, we unify these methods into a generalized, end-to-end parameter-efficient pre-training paradigm, which we refer to as ReLoRA* and formalize in Algorithm 1.

---
**Algorithm 1** ReLoRA*

---
**input** Initial weight $\mathbf{W}$ of each layer; total iteration $T_1$; iteration interval $T_2$
    **for** $t_1 = 0, \ldots, T_1 - 1$ **do**
        **Initializing:** Initialize $\mathbf{B}$ and $\mathbf{A}$ for each layer.
        **Subtracting:** Subtract $\mathbf{B}$ and $\mathbf{A}$ from $\mathbf{W}$ to maintain the original model output, $\mathbf{W} = \mathbf{W} - \mathbf{BA}$
        **Updating:** Update $\mathbf{B}$ and $\mathbf{A}$ for $T_2$ steps while keeping $\mathbf{W}$ frozen.
        **Merging:** Merge $\mathbf{B}$ and $\mathbf{A}$ back to $\mathbf{W}$, updating $\mathbf{W} = \mathbf{W} + \mathbf{BA}$.
    **end for**

---

For our experimental setup, ReLoRA* follows ReLoRA's initialization—$\mathbf{B}$ initialized to zero and $\mathbf{A}$ with a Kaiming initialization (He et al., 2015). The initial zero setting for $\mathbf{B}$ allows the subtraction step to be skipped. Notably, the optimizer states for $\mathbf{B}$ and $\mathbf{A}$ are reset after each merging step (99% optimizer state is pruned in ReLoRA).

**Limitation of ReLoRA*.** Each iteration of ReLoRA* learns only a small subset of singular values. Additionally, its reliance on zero initialization can result in zero gradients of low-rank matrices at each reinitialization, as discussed in Section 4.3. These issues hinder ReLoRA* from achieving the convergence speed and training quality of full-rank training.

### 3.3 GaLore

Gradient Low-rank Projection (GaLore) (Zhao et al., 2024) introduces a different approach by projecting the gradient using a low-rank projection matrix, rather than the weight matrix, as done by

LoRA and ReLoRA*. The projection matrix $\mathbf{P}_t$ is obtained by computing the top-$r$ singular vectors of the gradient of the weight matrix $\mathbf{W}$, and it is recalculated every $T$ steps. This matrix $\mathbf{P}_t$ is then used to project the gradient of the weight matrix into the low-rank space, allowing the low-rank gradient to update the first and second-order low-rank momentum in Adam. Finally, the low-rank updates calculated by Adam are re-projected back to the original weight shape and used to update the weights.

**Limitations of GaLore.**    Although GaLore presents a valuable contribution by exploring low-rank gradient projection, it has some limitations. Firstly, $\mathbf{P}_t$ is calculated based solely on the SVD of the gradient from a single batch, which can be affected by data sampling noise. Secondly, GaLore always selects the top-$r$ singular vectors, which, combined with the previous limitation, restricts its effectiveness during pre-training with a small $r$. In our experiments, we observed that with a small $r$ (less than $1/12$ of the dimension, different from the $1/2$ to $1/4$ used in the GaLore article), GaLore showed instability, leading to a sudden increase in loss on OPT-350M. Consequently, we chose to include the detailed explanation and comparison with GaLore in Appendix G rather than in the main text.

## 4    SPARSE SPECTRAL TRAINING

To address the limitations discussed previously, this section introduces Sparse Spectral Training (SST) and detailed its implementation.

### 4.1    SPARSE SPECTRAL LAYER

Sparse Spectral Training (SST) leverages sparse updates within the spectral domain of neural network weights. SST transforms each linear layer as follows:

$$\mathbf{h} = \mathbf{W}\mathbf{x} = \mathbf{U}\boldsymbol{\Sigma}\mathbf{V}^{\mathrm{T}}\mathbf{x}, \quad [\mathbf{U}, \boldsymbol{\Sigma}, \mathbf{V}^{\mathrm{T}}] = \mathrm{SVD}(\mathbf{W}) \qquad (3)$$

where $\mathbf{U} \in \mathbb{R}^{m \times m}$, $\boldsymbol{\Sigma} \in \mathbb{R}^{m \times m}$, and $\mathbf{V}^{\mathrm{T}} \in \mathbb{R}^{m \times n}$ represent the full-rank matrices derived from the singular value decomposition (SVD) of $\mathbf{W} \in \mathbb{R}^{m \times n}$, assuming $m \leq n$. It is important to note that unlike other LoRA-based methods, $\mathbf{U}, \boldsymbol{\Sigma}, \mathbf{V}^{\mathrm{T}}$ in this context are utilized at full rank, and the original weight matrix $\mathbf{W}$ is removed from networks. For simplicity, in the following discussion, we continue to use $\mathbf{W}$ to represent $\mathbf{U}\boldsymbol{\Sigma}\mathbf{V}^{\mathrm{T}}$.

The singular value decomposition is performed only during initialization and periodically reinitialized at each round (see Eq. 10), ensuring that the training process remains efficient (see

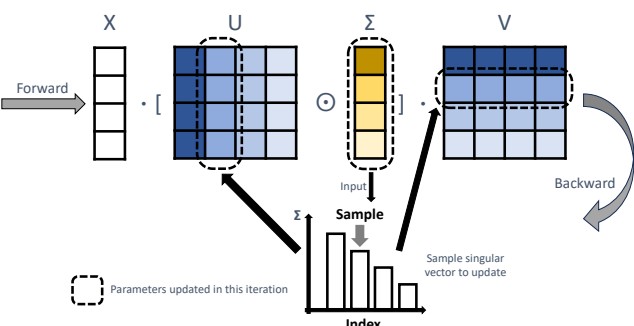

Figure 1: **Illustration of the Sparse Spectral Training (SST).** At each iteration, all singular values and selected singular vectors are updated based on their significance, determined by a multinomial sampling using singular values as probabilities.

Table 22 for the actual proportion of training time). However, as training progresses, $\mathbf{U}$, $\boldsymbol{\Sigma}$, and $\mathbf{V}^{\mathrm{T}}$ may gradually deviate from the true singular vectors and singular values of $\mathbf{W}$. In the subsequent section, we introduce improvements designed to mitigate this deviation.

### 4.2    GRADIENT UPDATE OF $\mathbf{U}, \mathbf{V}^{\mathrm{T}}$ WITH $\boldsymbol{\Sigma}$

**Update all $\boldsymbol{\Sigma}$.**    The diagonal matrix $\boldsymbol{\Sigma}$, simplified as a vector of dimension $m$, is updated at every step due to its low memory overhead. This ensures that all singular values are consistently adjusted to refine the model's performance. The update rule is as follows:

$$\Sigma^{t+1} = \max(\Sigma^t - \eta\nabla\mathcal{L}_{\Sigma}, 0) \tag{4}$$

where $\eta$ represents the learning rate, and $\nabla\mathcal{L}_{\Sigma}$ is the gradient backpropagated to $\Sigma$. The $\max$ function ensures that $\Sigma$ values remain non-negative.

**Selectively update U and $\mathbf{V}^{\mathrm{T}}$.** To update $\mathbf{U}$ and $\mathbf{V}^{\mathrm{T}}$, a selective updating strategy is employed, where specific parameters are chosen for each iteration based on a multinomial sampling method, as depicted in Figure 1. Consider $I = \{1, 2, ..., m\}$ as the set of all indices of singular vectors in $\mathbf{U}$ and $\mathbf{V}^{\mathrm{T}}$, with the sampling process defined by:

$$S \subseteq I, \quad S \sim \text{Multinomial}(r, \Sigma) \tag{5}$$

Here, $S$ represents the selected indices for update, with $|S| = r$, where $r$ is the predetermined number of vectors to be updated in each iteration. The update formulas for $\mathbf{U}$ and $\mathbf{V}^{\mathrm{T}}$ are:

$$\mathbf{U}^{t+1}_{\cdot i} = \mathbf{U}^t_{\cdot i} - \eta\nabla\mathcal{L}_{\mathbf{U}_{\cdot i}}, \quad \mathbf{V}^{t+1}_{\cdot i} = \mathbf{V}^t_{\cdot i} - \eta\nabla\mathcal{L}_{\mathbf{V}_{\cdot i}}, \quad \text{if } i \in S \tag{6}$$

where $\mathbf{U}_{\cdot i}$ means the $i$-th column vector of $\mathbf{U}$. To maintain the unit norm of each vector during training, and to ensure that magnitude information is encapsulated solely by $\Sigma$, the vectors are normalized post-update as follows:

$$\mathbf{U}^{t+1}_{\cdot i} = \frac{\mathbf{U}^t_{\cdot i} - \eta\nabla\mathcal{L}_{\mathbf{U}_{\cdot i}}}{|\mathbf{U}^t_{\cdot i} - \eta\nabla\mathcal{L}_{\mathbf{U}_{\cdot i}}|}, \quad \mathbf{V}^{t+1}_{\cdot i} = \frac{\mathbf{V}^t_{\cdot i} - \eta\nabla\mathcal{L}_{\mathbf{V}_{\cdot i}}}{|\mathbf{V}^t_{\cdot i} - \eta\nabla\mathcal{L}_{\mathbf{V}_{\cdot i}}|}, \quad \text{if } i \in S \tag{7}$$

**Enhance gradient of U and $\mathbf{V}^{\mathrm{T}}$.** Within a sparse spectral layer where $\mathbf{h} = \mathbf{U}\Sigma\mathbf{V}^{\mathrm{T}}\mathbf{x}$ (using $\mathbf{W}$ to denote $\mathbf{U}\Sigma\mathbf{V}^{\mathrm{T}}$), the gradients for $\mathbf{U}$ and $\mathbf{V}^{\mathrm{T}}$ are detailed below (derivation included in Appendix D):

$$\nabla\mathcal{L}_{\mathbf{U}_{\cdot i}} = \frac{\partial\mathcal{L}}{\partial\mathbf{U}_{\cdot i}} = \frac{\partial\mathcal{L}}{\partial\mathbf{W}}\mathbf{V}_{\cdot i}\Sigma_i, \quad \nabla\mathcal{L}_{\mathbf{V}_{\cdot i}} = \frac{\partial\mathcal{L}}{\partial\mathbf{V}_{\cdot i}} = \Sigma_i\frac{\partial\mathcal{L}}{\partial\mathbf{W^T}}\mathbf{U}_{\cdot i} \tag{8}$$

where $\mathbf{U}_{\cdot i}$ and $\mathbf{V}_{\cdot i}$ are column vectors of $\mathbf{U}$ and $\mathbf{V}^{\mathrm{T}}$, respectively, and $\Sigma_i$ represents the diagonal elements of $\Sigma$. This represents the default gradient calculation for these matrices. We propose an enhanced gradient calculation for $\mathbf{U}_{\cdot i}$ and $\mathbf{V}_{\cdot i}$ as follows:

$$\tilde{\nabla}\mathcal{L}_{\mathbf{U}_{\cdot i}} = \frac{\partial\mathcal{L}}{\partial\mathbf{W}}\mathbf{V}_{\cdot i}, \quad \tilde{\nabla}\mathcal{L}_{\mathbf{V}_{\cdot i}} = \frac{\partial\mathcal{L}}{\partial\mathbf{W^T}}\mathbf{U}_{\cdot i} \tag{9}$$

In the enhanced gradient, the learning of direction ($\mathbf{U}_{\cdot i}$ and $\mathbf{V}_{\cdot i}$) is decoupled from the magnitude ($\Sigma_i$), allowing singular vectors with lower singular values to retain substantial gradients.

**Periodic re-SVD.** During training, the orthogonality among the vectors of $\mathbf{U}$ and $\mathbf{V}^{\mathrm{T}}$ tends to diminish. Preserving the orthogonality of these singular vectors is crucial, as it prevents the learning process from degenerating into a low-rank subspace, thus preserving the model's full expressive capabilities. To maintain this orthogonality, it is essential to periodically perform singular value decomposition:

$$[\mathbf{U}^{t+1}, \Sigma^{t+1}, \mathbf{V}^{t+1^{\mathrm{T}}}] = \text{SVD}(\mathbf{U}^t\Sigma^t\mathbf{V}^{t^{\mathrm{T}}}) \tag{10}$$

Each time we perform this re-SVD, we consider it a new **round**. Each time we select vectors for updating, as described in Eq. 5, we call it a new **iteration**. The full method is detailed in Algorithm 2.

## 4.3 WHY SVD DECOMPOSITION IS IMPORTANT

This section discusses the advantages of using SVD initialization and periodic re-SVD over zero initialization as employed in ReLoRA* methods.

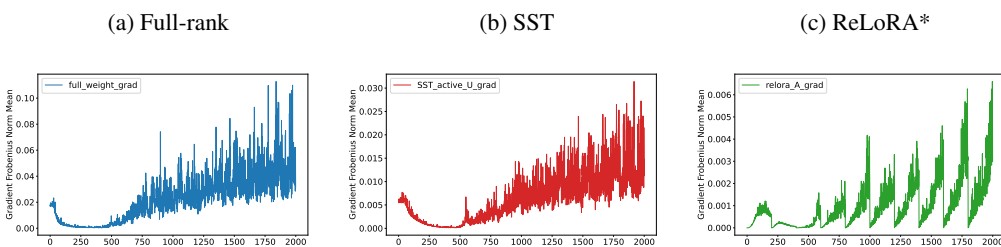

Figure 2: **ReLoRA\* suffers saddle point issue at each restart.** This plot depicts the average Frobenius Norm of gradients of: (a) all weight matrices in full-rank training; (b) all sampled $\mathbf{U}$ in SST; (c) all $\mathbf{A}$ in ReLoRA\*, in first 2000 steps. All methods are trained on Transformer with dimension $= 64$, $r = 8$ on IWSLT'14. Both SST and ReLoRA\* set iteration interval to 200. When the average Frobenius Norm of gradients approaches zero, it indicates that a saddle point issue happens. Figure (c) shows that ReLoRA\* suffers saddle point issue periodically at the beginning of each iteration. The correlation between SST and full-rank gradient norm along the steps is **0.85**, whereas the correlation between ReLoRA\* and full-rank is **0.58**. This demonstrates that the gradient curve of SST more closely approximate the gradient curve of full-rank training, compared with ReLoRA\*.

**Saddle point issues after each merging in ReLoRA\*.** The gradient of $\mathbf{A}$ and $\mathbf{B}$ in ReLoRA\* is:

$$\frac{\partial \mathcal{L}}{\partial \mathbf{B}} = \frac{\partial \mathcal{L}}{\partial \mathbf{W}} \mathbf{A}^{\mathrm{T}} \quad \text{and} \quad \frac{\partial \mathcal{L}}{\partial \mathbf{A}} = \mathbf{B}^{\mathrm{T}} \frac{\partial \mathcal{L}}{\partial \mathbf{W}} \tag{11}$$

After each merging, $\mathbf{B}$ is reinitialized to zero, and the gradient of $\mathbf{A}$ is calculated as $\frac{\partial \mathcal{L}}{\partial \mathbf{A}} = \mathbf{0}^{\mathrm{T}} \frac{\partial \mathcal{L}}{\partial \mathbf{W}} = \mathbf{0}$, which causes a slow learning progress at the beginning of each iteration. Additionally, in ReLoRA\*, resetting the momentum of $\mathbf{B}$ and $\mathbf{A}$ after each merging aggravates this issue, particularly when the merging interval $T_2$ is short.

**Compared with ReLoRA\*, SST more closely approximates full-rank training.** In Figure 2, we compare the average Frobenius Norm of gradients of weight matrices in full-rank training, low rank matrices in SST and ReLoRA\*. This plot shows that ReLoRA\* suffers saddle point issue periodically at the beginning of each iteration. We also calculate the correlation between SST and full-rank gradient norm along the steps is **0.85**, whereas the correlation between ReLoRA\* and full-rank is **0.58**. The fact that SST's gradient norm is more closely correlated with the full-rank gradient norm than ReLoRA\* suggests that SST more closely approximates the gradient of full-rank training.

SST initializes and reinitializes its low-rank matrices $\mathbf{U}$ and $\mathbf{V}$ using the singular vectors of $\mathbf{W}$. In contrast to ReLoRA\*, which relies on random or zero initialization for its low-rank matrices, SST better captures the direction of $\mathbf{W}$'s updates, allowing it to more closely approximate full-rank training. As demonstrated in the ablation study (Appendix H), replacing SVD-based initialization with random initialization leads to a significant drop in performance, highlighting the critical role of SVD in SST's effectiveness.

### 4.4 SST BALANCES EXPLOITATION AND EXPLORATION

From another perspective, SST combines the strategies of exploitation and exploration in spectral domain. LoRA primarily focuses on exploitation by repeatedly adjusting the top-$r$ singular values, as detailed in Section 3.1, while neglecting the remaining spectral vectors. ReLoRA\*, on the other hand, emphasizes exploration by periodically reinitializing the matrices $\mathbf{B}$ and $\mathbf{A}$ after each merging, thereby constantly seeking new directions for learning but ignoring previously established dominant directions.

SST boosts learning efficiency by updating all magnitudes ($\mathbf{\Sigma}$) at each step and cyclically revisiting previously established dominant directions. By continuously updating all singular values, SST ensures unbiased sampling of $\mathbf{U}$ and $\mathbf{V}^{\mathrm{T}}$, enabling a thorough exploration of the parameter space. As

a result, SST balances the exploitation of known critical directions with the exploration of emerging opportunities within the spectrum of matrix decomposition.

### 4.5 SPARSITY OF SST

We analyze the efficiency of parameter usage.. Specifically, the ratio of trainable parameters in SST at a given rank $r$, denoted as $\Gamma_{\text{SST},r}$, is calculated as $\frac{r(m+n)+m}{mn}$. This parameter ratio is slightly higher than that of LoRA at the same rank, $\Gamma_{\text{LoRA},r} = \frac{r(m+n)}{mn}$, yet remains lower than LoRA at rank $r+1$, $\Gamma_{\text{LoRA},r+1} = \frac{(r+1)(m+n)}{mn}$, indicating a slightly increase in trainable parameters.

### 4.6 MEMORY-EFFICIENT IMPLEMENTATION FOR SST

To achieve similar memory reduction as LoRA, SST stores optimizer states for all $\boldsymbol{\Sigma}$ and only for the vectors sampled in each iteration from $\mathbf{U}$ and $\mathbf{V}^{\text{T}}$. However, standard implementations of Adam optimizer (Kingma & Ba, 2014) in PyTorch (Paszke et al., 2019) do not support sparse optimizer states. To address this, we partition $\mathbf{U}$ and $\mathbf{V}^{\text{T}}$ into active and frozen segments. Only active segments store the optimizer states, where $\mathbf{U}_{\text{active}} \in \mathbb{R}^{m \times r}$ and $\mathbf{V}^{\text{T}}_{\text{active}} \in \mathbb{R}^{r \times n}$. The frozen segments, $\mathbf{U}_{\text{freeze}}$ and $\mathbf{V}^{\text{T}}_{\text{freeze}}$, do not store optimizer states. Vectors newly sampled from the frozen segments are swapped with unsampled vectors in the active segments (illustrated in Figure 3). This approach enables SST to func-

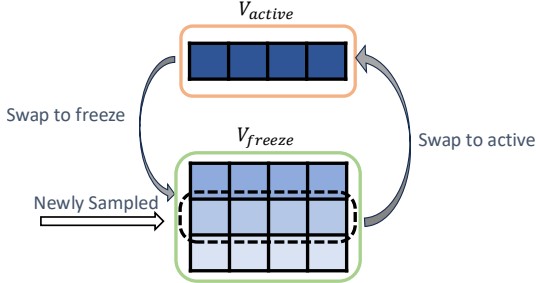

Figure 3: **Illustration of the memory-efficient implementation for SST.** After each sampling step, the sampled vectors are swapped with the active vectors from the previous iteration.

tion as a time-sharing operating system, effectively balancing resource allocation among the vectors in $\mathbf{U}$ and $\mathbf{V}^{\text{T}}$.

## 5 EXPERIMENTS

To validate our Sparse Spectral Training (SST) approach, we conducted experiments on both Euclidean and hyperbolic neural networks, demonstrating the generalization of SST across various neural network architectures and embedding geometries.

We compared SST with full-rank training, LoRA, and ReLoRA*. The key distinctions between ReLoRA* and ReLoRA (Lialin et al., 2024) is that ReLoRA includes a full-rank training as "warm start", which prevents it from being an end-to-end memory-efficient pre-training method.

For all low-rank methods, all linear layers in the baseline models were replaced by low-rank layers. Hyperparameters and implementation details are provided in Appendix E.

As discussed in Section 3.3, the comparison between SST and the contemporaneous work GaLore (Zhao et al., 2024) is provided in Appendix G, as GaLore is unstable during OPT pre-training with $r = 64$. We highlight SST's superior performance across all of our experiment settings. Ablation studies are documented in Appendix H, and a detailed analysis of memory consumption and training time can be found in Appendix I. Additionally, an experiment on image classification tasks is included in Appendix J.

### 5.1 MACHINE TRANSLATION

We employ the vanilla transformer (Vaswani et al., 2017) as the Euclidean transformer and HyboNet (Chen et al., 2022) as the hyperbolic transformer. Our experiments include three widely-used machine translation datasets: IWSLT'14 English-to-German (Cettolo et al., 2014), IWSLT'17 German-to-English (Cettolo et al., 2017), and Multi30K German-to-English (Elliott et al., 2016). For IWSLT'14, the hyperparameters are aligned with those from HyboNet.

Table 1: **BLEU scores on IWSLT'14 for Euclidean and hyperbolic Transformers.** Values in bold indicate the highest performance among low-rank methods. Values marked with an "*" exceed the performance of their full-rank counterparts. Some BLEU scores are zero because that training resulted in NaN losses. Notably, SST consistently outperforms other low-rank methods. Furthermore, the hyperbolic Transformer trained by SST shows improved performance over the full-rank hyperbolic Transformer, particularly as the dimension size increases.

| Dimension | r | Euclidean | | | | Hyperbolic | | | |
|---|---|---|---|---|---|---|---|---|---|
| | | Full | LoRA | ReLoRA* | SST | Full | LoRA | ReLoRA* | SST |
| 64 | 8 | 24.27 | 18.08 | 18.12 | **22.28** | 25.69 | 17.50 | 0.0 | **23.40** |
| | 4 | | 14.05 | 15.49 | **20.27** | | 0.0 | 0.0 | **23.03** |
| 128 | 16 | 25.79 | 23.30 | 22.92 | **25.12** | 24.70 | 23.70 | 0.0 | **25.22*** |
| | 8 | | 20.56 | 20.61 | **24.19** | | 20.81 | 0.0 | **25.12*** |
| | 4 | | 16.37 | 18.00 | **22.80** | | 17.58 | 24.42 | **24.60** |
| 256 | 32 | 23.92 | 23.76 | 23.02 | **23.97*** | 19.94 | 24.16* | 0.0 | **25.04*** |
| | 16 | | 22.88 | 22.01 | **23.42** | | 23.93* | 0.0 | **25.52*** |
| | 8 | | 20.32 | 20.36 | **22.65** | | 21.58* | 24.02* | **24.67*** |
| | 4 | | 16.72 | 17.85 | **21.39** | | 18.72 | 24.08* | **24.51*** |

**Euclidean Transformer** Table 1 presents BLEU scores for IWSLT'14 across various dimensions and ranks ($r$). The results confirm that SST consistently outperforms other low-rank methods. On average, SST reduces the BLEU gap (defined as the BLEU score difference from full-rank training) by **66.7%** for Euclidean Transformers on IWSLT'14.

Further comparative results on the Multi30K and IWSLT'17 datasets using the standard di-

Table 2: **Comparison of BLEU scores on Multi30k and IWSLT'17 datasets** using Euclidean Transformer (dimension = 512), $r = 32$. Scores highlighted in bold represent the highest performance achieved by low-rank methods.

| | Full | LoRA | ReLoRA* | SST |
|---|---|---|---|---|
| **Multi30K** | 40.7 | 40.1 | 41.6 | **43.4** |
| **IWSLT'17** | 31.7 | 31.9 | 32.0 | **32.3** |

mensions for vanilla Euclidean transformers are documented in Table 2. Here, SST not only surpasses other low-rank methods but also demonstrates superior performance compared to full-rank training.

**Hyperbolic Transformer** In Table 1, some BLEU scores for the hyperbolic transformer are zero, due to the training process encountering NaN losses, whereas SST maintains stability throughout. SST consistently outperforms other low-rank methods across all settings and even exceeds the performance of full-rank training in various configurations.

Previous hyperbolic neural network articles have predominantly focused on low-dimensional configurations (Ganea et al., 2018; Shimizu et al., 2021; Nickel & Kiela, 2017). A key characteristic of hyperbolic space is its exponential growth in volume with distance from a reference point, which is significantly more rapid than the polynomial growth seen in Euclidean space (Cho et al., 2019). This expansive nature makes hyperbolic spaces particularly prone to overfitting as dimensionality increases. By imposing constraints on the parameter search space of hyperbolic neural networks, SST prevents the overfitting typically associated with such high-dimensional settings.

## 5.2 NATURAL LANGUAGE GENERATION

**Language modeling.** We utilize the OPT (Zhang et al., 2022) and LLaMA (Touvron et al., 2023a) architecture as the baseline for our language generation experiments. For LLaMA, we follow the experiment setup from (Zhao et al., 2024). All models are pre-trained on OpenWebText (Gokaslan & Cohen, 2019), an open-source reproduction of OpenAI's WebText. We applied a rank of $r = 64$ for all OPT models and LLaMA-130M, and $r = 128$ for LLaMA-1.3B.

Table 3 displays the validation perplexity results on the OpenWebText dataset across different sizes of all LLMs. The results indicate that SST achieves lower perplexity scores compared to LoRA and ReLoRA*, significantly reducing the perplexity gap—defined as the difference between the

Table 3: **Validation perplexity on OpenWebText** across various model sizes of OPT and LLaMA along with the number of trainable parameters of each method. Values in bold highlight the highest performance among the low-rank methods.

| Model | $r/d_{\text{model}}$ | Training Tokens | Full | LoRA | ReLoRA* | SST |
|---|---|---|---|---|---|---|
| OPT-125M | 64/768 | 19.7B | 23.50 (125.2M) | 34.23 (50.9M) | 35.80 (50.9M) | **26.98** (51.0M) |
| OPT-350M | 64/1024 | 19.7B | 21.78 (331.2M) | 34.26 (57.5M) | 39.21 (57.5M) | **27.72** (57.7M) |
| OPT-1.3B | 64/2048 | 19.7B | 15.10 (1.316B) | 1716 (164.4M) | 29.52 (164.4M) | **22.31** (164.7M) |
| LLaMA-130M | 64/768 | 2.6B | 20.04 (134.11M) | 29.71 (60.38M) | 31.33 (60.38M) | **23.35** (60.44M) |
| LLaMA-1.3B | 128/2048 | 13.1B | 14.54 (1.339B) | 16.50 (250.71M) | 17.32 (250.71M) | **14.59** (251.05M) |

perplexity of the low-rank method and the full-rank training. Specifically, SST reduces this gap by **67.6%** (OPT-125M), **52.4%** (OPT-350M), **50.0%** (OPT-1.3B), **65.8%** (LLaMA-130M), and **97.4%** (LLaMA-1.3B).

Figure 4 presents a plot of validation loss against effective steps for various training methods. The effective step metric, defined as the product of the number of training steps and the number of trainable parameters, provides insight into the efficiency of parameter updates. Although parameter-efficient training methods typically exhibit slower convergence compared to full-rank training, the effective step metric illustrates that SST updates parameters more effectively. At the final effective step for SST on OPT-1.3B, SST achieves a validation perplexity of **22.31**, whereas full-rank training at the same effective step only reaches a validation perplexity of **34.05**, demonstrating that SST is more efficient in updating parameters compared to full-rank training.

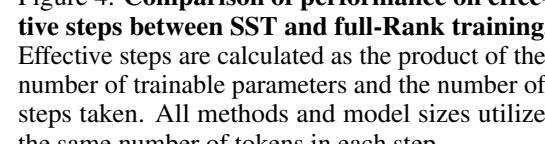

Figure 4: **Comparison of performance on effective steps between SST and full-Rank training.** Effective steps are calculated as the product of the number of trainable parameters and the number of steps taken. All methods and model sizes utilize the same number of tokens in each step.

**Zero-shot evaluations.** Each pretrained model performs zero-shot evaluations on all 16 NLP tasks used in the OPT article (Zhang et al., 2022), including ARC Easy and Challenge (Clark et al., 2018), HellaSwag (Zellers et al., 2019), OpenBookQA (Mihaylov et al., 2018), PIQA (Bisk et al., 2020), StoryCloze (Mostafazadeh et al., 2016), SuperGLUE (Wang et al., 2019), WinoGrad (Levesque et al., 2012), and WinoGrande (Sakaguchi et al., 2019). Evaluations are conducted using the LM Evaluation Harness framework (Gao et al., 2023). Except for the ReCoRD task, which uses F1 score, all other tasks are evaluated using accuracy.

Table 4 presents the zero-shot evaluation results across the 16 NLP tasks. SST achieves a higher average score than other low-rank methods across all sizes of the OPT models. On the OPT-125M, the average score for zero-shot evaluations of SST is **44.6**, slightly exceeding the average score of full-rank training, which is **44.5**. Additionally, we calculated the win percentage (including ties) for each low-rank method compared to full-rank training. On the OPT-125M, the win percentage of SST is **56.3%**, indicating that SST performed as well as or better than full-rank training on more than half of the zero-shot evaluation tasks.

## 5.3 HYPERBOLIC GRAPH NEURAL NETWORKS

Hyperbolic Graph Neural Networks (HGNNs) (Chami et al., 2019; Chen et al., 2022) capitalize on the expansive and hierarchical nature of hyperbolic space to efficiently manage and analyze graph-structured data. This geometric space is particularly suitable for graphs due to its ability to closely mimic the underlying data structures with minimal distortion, offering a substantial improvement over traditional Euclidean methods.

We evaluated the effectiveness of SST on HyboNet (Chen et al., 2022) version of HGNN in node classification and link prediction across four distinct datasets: Airport (Chami et al., 2019), Cora (Sen

Table 4: **Zero-shot evaluations** on the same 16 NLP tasks featured in the OPT article (Zhang et al., 2022). Except for the ReCoRD task, which uses F1 score, all other tasks are evaluated using accuracy, with values presented as percentages. Mean scores in bold represent superior performance among the low-rank methods. Additionally, we include the win percentage (including ties) for each low-rank method compared to the full-rank training.

| | OPT-125M | | | | OPT-350M | | | | OPT-1.3B | | | |
|---|---|---|---|---|---|---|---|---|---|---|---|---|
| | **Full** | **LoRA** | **ReLoRA*** | **SST** | **Full** | **LoRA** | **ReLoRA*** | **SST** | **Full** | **LoRA** | **ReLoRA*** | **SST** |
| ARC (Challenge) | 21.2 | 22.9 | 21.1 | 21.3 | 22.0 | 22.3 | 21.3 | 21.1 | 24.6 | 24.2 | 22.9 | 21.5 |
| ARC (Easy) | 35.8 | 34.2 | 33.9 | 34.3 | 35.9 | 32.3 | 33.0 | 35.7 | 43.2 | 26.1 | 35.9 | 37.8 |
| BoolQ | 59.5 | 54.2 | 60.8 | 62.0 | 53.6 | 56.2 | 62.2 | 57.7 | 57.7 | 37.8 | 61.4 | 59.5 |
| CB | 51.8 | 48.2 | 28.6 | 48.2 | 44.6 | 44.6 | 33.9 | 41.1 | 59.0 | 41.1 | 37.5 | 42.9 |
| COPA | 67.0 | 61.0 | 57.0 | 66.0 | 69.0 | 61.0 | 59.0 | 60.0 | 70.0 | 51.0 | 68.0 | 65.0 |
| HellaSwag | 27.7 | 26.5 | 27.1 | 26.9 | 28.4 | 26.6 | 26.9 | 27.5 | 35.0 | 26.1 | 27.2 | 28.1 |
| MultiRC | 55.4 | 57.2 | 55.9 | 57.2 | 52.0 | 52.6 | 56.4 | 57.0 | 56.8 | 42.8 | 57.7 | 56.9 |
| OpenBookQA | 24.6 | 24.6 | 23.6 | 26.2 | 26.4 | 24.2 | 23.0 | 25.2 | 29.0 | 27.0 | 24.8 | 25.0 |
| PIQA | 58.7 | 57.2 | 56.3 | 58.3 | 59.2 | 56.9 | 56.9 | 59.0 | 64.0 | 50.3 | 57.1 | 59.1 |
| ReCoRD | 16.7 | 17.5 | 22.6 | 18.5 | 19.4 | 17.6 | 19.0 | 23.2 | 13.7 | 17.6 | 23.0 | 18.1 |
| RTE | 50.5 | 56.7 | 53.1 | 53.4 | 52.0 | 49.1 | 54.9 | 50.2 | 51.6 | 52.7 | 52.0 | 53.8 |
| StoryCloze | 55.8 | 53.8 | 53.6 | 54.5 | 57.2 | 53.7 | 53.0 | 54.6 | 61.1 | 49.7 | 54.0 | 56.1 |
| WIC | 49.8 | 51.4 | 50.0 | 50.0 | 50.5 | 50.0 | 50.0 | 50.2 | 50.3 | 50.0 | 50.0 | 50.0 |
| Winograd | 52.0 | 48.7 | 50.6 | 50.6 | 55.0 | 51.7 | 50.2 | 51.3 | 55.7 | 50.9 | 52.4 | 55.3 |
| Winogrande | 49.1 | 49.2 | 50.7 | 50.1 | 50.7 | 50.3 | 50.8 | 52.0 | 51.1 | 47.9 | 50.0 | 49.1 |
| WSC | 36.5 | 38.5 | 36.5 | 36.5 | 36.5 | 37.5 | 36.5 | 36.5 | 39.4 | 63.5 | 36.5 | 36.5 |
| Mean | 44.5 | 43.8 | 42.6 | **44.6** | 44.5 | 42.9 | 42.9 | **43.9** | 47.6 | 41.2 | 44.4 | **44.7** |
| Win Percentage | - | 50.0 | 43.8 | 56.3 | - | 31.3 | 31.3 | 31.3 | - | 18.8 | 25.0 | 25.0 |

Table 5: **Node Classification and Link Prediction Results.** Model's dimension $d = 16$. Results are reported as test F1 scores for node classification and test precision for link prediction, expressed in percentages. Values highlighted in bold represent the highest performance among the low-rank methods, while those marked with an "*" denote performance that exceeds that of the full-rank variants.

| | Node Classification | | | | Link Prediction | | | |
|---|---|---|---|---|---|---|---|---|
| **Method** | **Airport** | **Cora** | **Disease** | **PubMed** | **Airport** | **Cora** | **Disease** | **PubMed** |
| **Full** $d = 16$ | 92.88 ± 0.5 | 81.13 ± 0.2 | 91.83 ± 0.4 | 78.1 ± 0.4 | 95.77 ± 0.08 | 94.62 ± 0.2 | 91.49 ± 1.5 | 96.55 ± 0.03 |
| **LoRA** $r = 1$ | 85.75 ± 1.0 | 45.5 ± 0.3 | 79.66 ± 1.9 | 69.17 ± 2.1 | 94.01 ± 0.2 | 84.22 ± 0.1 | 84.29 ± 1.5 | 89.34 ± 0.4 |
| **SST** $r = 1$ | **88.61 ± 0.5** | **75.07 ± 0.5** | **89.22 ± 1.7** | **77.47 ± 0.3** | **95.37 ± 0.4** | **91.11 ± 0.6** | **93.63 ± 0.7*** | **95.57 ± 0.1** |
| **LoRA** $r = 2$ | **89.06 ± 1.0** | 64.73 ± 0.8 | 83.84 ± 4.3 | 76.27 ± 0.8 | 94.75 ± 0.15 | 88.8 ± 0.5 | 91.38 ± 0.7 | 92.14 ± 0.3 |
| **SST** $r = 2$ | 87.92 ± 0.09 | **77.5 ± 0.7** | **90.64 ± 1.7** | **77.93 ± 0.1** | **95.59 ± 0.2** | **91.89 ± 0.3** | **94.83 ± 0.6*** | **95.71 ± 0.1** |

et al., 2008), Disease (Anderson & May, 1991), and PubMed (Namata et al., 2012). Each experiment was conducted with three random seeds.

The results, detailed in Table 5, demonstrate SST has strong performance in both node classification and link prediction tasks. With $r = 1$, SST reduces the performance gap, by an average of **73.7%** in node classification and **82.5%** in link prediction. In the Disease link prediction task, SST outperforms full-rank training at both $r = 1$ and $r = 2$. Notably, SST's advantage over LoRA is greater at $r = 1$ than at $r = 2$, likely due to SST's sampling strategy being particularly effective in sparser scenarios.

## 6 CONCLUSION AND DISCUSSION

In this work, Sparse Spectral Training (SST) has demonstrated its efficacy as a parameter-efficient pre-training methodology that surpasses other parameter-efficient methods, and better approximates the learning dynamics and performance of full-rank training across diverse architectures, tasks, and embedding geometries. SST introduces a novel approach by updating all singular values and selectively adjusting the singular vectors of network weights. Moreover, SST incorporates SVD both for the initialization and periodic reinitialization of low-rank parameters. Future directions for SST include: (1) Investigating faster convergence approaches that avoid optimizer state reset. (2) Extending the application of SST to the embeddings of large language models (LLMs).

**Reproducibility Statement.** To facilitate reproducibility, we provide the source code for Sparse Spectral Training (SST), along with detailed instructions for running experiments, at `https://anonymous.4open.science/r/sparse_spectral_training-6A2C/`. All hyper-parameters, model architectures, and training settings for all methods are documented in Appendix E. These resources are intended to provide all the necessary information for reproducing our results.

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

# A  ALGORITHM OF SPARSE SPECTRAL TRAINING

---

**Algorithm 2** Sparse Spectral Training (SST)

---

**input** Dataset $D$; total round $T_1$; number of iterations $T_2$; iteration interval $T_3$

Use Kaiming initialization to initialize origin model's weight $\mathbf{W}_k^{(0)}$, $k = 1, ..., n$, where $n$ is the number of linear layers.

Replace origin model's weight with SVD decomposition

$$[\mathbf{U}_k^{(t_1,0)}, \mathbf{\Sigma}_k^{(t_1,0)}, \mathbf{V}_k^{(t_1,0)\,\mathrm{T}}] = \mathrm{SVD}(\mathbf{W}_k^{(t_1)})$$

**for** $t_1 = 0, \ldots, T_1 - 1$ **do**

    **for** $t_2 = 0, \ldots, T_2 - 1$ **do**

        $I_k = \{1, 2, \ldots, m\}$ be the set of all possible indices of singular vectors

$$S_k^{(t_1,t_2)} \subseteq I_k, \quad S_k^{(t_1,t_2)} \sim \mathrm{Multinomial}(r, \mathbf{\Sigma}_k^{(t_1, t_2 \times T_3)})$$

        **for** $t_3 = 0, \ldots, T_3 - 1$ **do**

            Represent $t = t_2 \times T_3 + t_3$;

            Sample a mini-batch from $D$ and compute the forward pass by Eq.3 and compute the gradient $\nabla \mathcal{L}$;

            Update $\mathbf{\Sigma}_k^{(t_1,t+1)} = \max(\mathbf{\Sigma}_k^{(t_1,t)} - \eta \nabla \mathcal{L}_{\mathbf{\Sigma}_k}, 0)$

            Update

$$\mathbf{U}_{k,\cdot i}^{(t_1,t+1)} = \frac{\mathbf{U}_{k,\cdot i}^{(t_1,t)} - \eta \tilde{\nabla} \mathcal{L}_{\mathbf{U}_{k,\cdot i}}}{|\mathbf{U}_{k,\cdot i}^{(t_1,t)} - \eta \tilde{\nabla} \mathcal{L}_{\mathbf{U}_{k,\cdot i}}|}, \quad \mathbf{V}_{k,\cdot i}^{(t_1,t+1)} = \frac{\mathbf{V}_{k,\cdot i}^{(t_1,t)} - \eta \tilde{\nabla} \mathcal{L}_{\mathbf{V}_{k,\cdot i}}}{|\mathbf{V}_{k,\cdot i}^{(t_1,t)} - \eta \tilde{\nabla} \mathcal{L}_{\mathbf{V}_{k,\cdot i}}|}, \quad \text{if } i \in S_k^{(t_1,t_2)}$$

            where $\mathbf{U}_{k,\cdot i}$ means column vector i of $\mathbf{U}_k$

        **end for**

    **end for**

    Reinitialize with new SVD decomposition

$$[\mathbf{U}_k^{(t_1+1,0)}, \mathbf{\Sigma}_k^{(t_1+1,0)}, \mathbf{V}_k^{(t_1+1,0)\,\mathrm{T}}] = \mathrm{SVD}(\mathbf{U}_k^{(t_1, T_2 \times T_3 - 1)} \mathbf{\Sigma}_k^{(t_1, T_2 \times T_3 - 1)} \mathbf{V}_k^{(t_1, T_2 \times T_3 - 1)\,\mathrm{T}})$$

**end for**

---

# B  EXPERIMENTS ON LARGER DATASETS AND HYPERPARAMETER TUNING

To further evaluate the performance of SST, we conducted additional experiments using larger datasets and varied hyperparameter settings. Specifically, we pre-trained LLaMA-130M on the C4 dataset (Raffel et al., 2020), which is about 25 times larger than OpenWebText. We also compared the performance of SST, LoRA, and ReLoRA* under two different learning rates.

Table 6 presents the validation perplexity (PPL) results for LLaMA-130M on both C4 and OpenWebText. The results show that SST consistently outperforms other low-rank methods, achieving lower perplexity across all configurations.

Table 6: **Validation perplexity on C4 and OpenWebText** for LLaMA-130M with different learning rates. Bold values indicate the lowest PPL among all low-rank methods.

| Dataset | Model | $r/d$ | Full (lr=1e-3) | lr=1e-3 | | | lr=3e-3 | | |
|---|---|---|---|---|---|---|---|---|---|
| | | | | LoRA | ReLoRA* | SST | LoRA | ReLoRA* | SST |
| C4 | LLaMA-130M | 64/768 | 24.91 | 35.91 | 37.34 | 25.89 | 32.13 | 30.75 | 133.06 | **29.79** |
| OpenWebText | LLaMA-130M | 64/768 | 20.04 | 29.71 | 31.33 | 25.89 | 795.24 | 230.43 | **23.35** |

Each method was trained with 2.6 billion tokens. The learning rate of $1\mathrm{e}{-3}$ for full-rank training aligns with the configuration used in the ReLoRA article. For consistency, we applied the same learning rates ($lr = 1\mathrm{e}{-3}$ and $lr = 3\mathrm{e}{-3}$) across LoRA, ReLoRA*, and SST.

SST consistently achieves lower perplexity than LoRA and ReLoRA* at the same learning rate. Notably, with $lr = 3\mathrm{e}{-}3$, SST surpasses all other low-rank methods, reducing the perplexity gap by **16.4%** on C4 and **65.8%** on OpenWebText. These findings highlight SST's effectiveness and robustness on larger datasets and varied learning rate configurations.

## C RELATED WORK OF OTHER PARAMETER-EFFICIENT TRAINING METHODS

Apart from low-rank adaptations, researchers have developed a variety of parameter-efficient training techniques to optimize resource consumption while preserving learning effectiveness. Prompt tuning is an effective method that integrates tunable prefixes or soft prompts into the input embeddings of models. It enables lightweight task-specific adaptations with minimal impact on the model's overall architecture (Lester et al., 2021; Liu et al., 2021). Dynamic sparse training (DST), through methods like SET (Mocanu et al., 2018), RIGL (Evci et al., 2020), MEST (Yuan et al., 2021), and CHT (Zhang et al., 2024), employs a dynamic prune-and-grow strategy that adjusts network topology during training. This approach optimizes training efficiency and can improve generalization by continuously adapting the network's sparse structure. This presents a significant shift from static training methods.

## D PROOF OF GRADIENT OF SPARSE SPECTRAL LAYER

We can express the differential of $\mathbf{W}$ as the sum of differentials:

$$d\mathbf{W} = d\mathbf{U}\,\boldsymbol{\Sigma}\mathbf{V}^{\mathrm{T}} + \mathbf{U}\,d\boldsymbol{\Sigma}\,\mathbf{V}^{\mathrm{T}} + \mathbf{U}\boldsymbol{\Sigma}\,d\mathbf{V}^{\mathrm{T}} \tag{12}$$

We have chain rule for the gradient of $\mathbf{W}$:

$$\frac{\partial \mathcal{L}}{\partial \mathbf{W}} = \frac{\partial \mathcal{L}}{\partial \mathbf{h}}\frac{\partial \mathbf{h}}{\partial \mathbf{W}} = \frac{\partial \mathcal{L}}{\partial \mathbf{h}}\mathbf{x}^{\mathrm{T}} \tag{13}$$

$$\begin{aligned}
d\mathcal{L} &= \frac{\partial \mathcal{L}}{\partial \mathbf{W}} : d\mathbf{W} \\
&= \frac{\partial \mathcal{L}}{\partial \mathbf{W}} : d\mathbf{U}\,\boldsymbol{\Sigma}\mathbf{V}^{\mathrm{T}} + \frac{\partial \mathcal{L}}{\partial \mathbf{W}} : \mathbf{U}\,d\boldsymbol{\Sigma}\,\mathbf{V}^{\mathrm{T}} + \frac{\partial \mathcal{L}}{\partial \mathbf{W}} : \mathbf{U}\boldsymbol{\Sigma}\,d\mathbf{V}^{\mathrm{T}} \\
&= \frac{\partial \mathcal{L}}{\partial \mathbf{W}}\mathbf{V}\boldsymbol{\Sigma} : d\mathbf{U} + \mathbf{U}^{\mathrm{T}}\frac{\partial \mathcal{L}}{\partial \mathbf{W}}\mathbf{V} : d\boldsymbol{\Sigma} + \boldsymbol{\Sigma}\mathbf{U}^{\mathrm{T}}\frac{\partial \mathcal{L}}{\partial \mathbf{W}} : d\mathbf{V}^{\mathrm{T}}
\end{aligned}$$

where : is the Frobenius inner product. So we have the gradient of $\mathbf{U}$, $\boldsymbol{\Sigma}$ and $\mathbf{V}^{\mathrm{T}}$:

$$\frac{\partial \mathcal{L}}{\partial \mathbf{U}} = \frac{\partial \mathcal{L}}{\partial \mathbf{W}}\mathbf{V}\boldsymbol{\Sigma}, \quad \frac{\partial \mathcal{L}}{\partial \mathbf{V}^{\mathrm{T}}} = \boldsymbol{\Sigma}\mathbf{U}^{\mathrm{T}}\frac{\partial \mathcal{L}}{\partial \mathbf{W}}, \quad \frac{\partial \mathcal{L}}{\partial \boldsymbol{\Sigma}} = \mathbf{U}^{\mathrm{T}}\frac{\partial \mathcal{L}}{\partial \mathbf{W}}\mathbf{V} \tag{14}$$

In vector perspective, for the $i^{th}$ vector, it is:

$$\frac{\partial \mathcal{L}}{\partial \mathbf{U}_{\cdot i}} = \frac{\partial \mathcal{L}}{\partial \mathbf{W}}\mathbf{V}_{\cdot i}\boldsymbol{\Sigma}_i, \quad \frac{\partial \mathcal{L}}{\partial \mathbf{V}_{\cdot i}} = \boldsymbol{\Sigma}_i\frac{\partial \mathcal{L}}{\partial \mathbf{W}^{\mathrm{T}}}\mathbf{U}_{\cdot i}, \quad \frac{\partial \mathcal{L}}{\partial \boldsymbol{\Sigma}_i} = \mathbf{U}_{\cdot i}{}^{\mathrm{T}}\frac{\partial \mathcal{L}}{\partial \mathbf{W}}\mathbf{V}_{\cdot i} \tag{15}$$

where $\mathbf{U}_{\cdot i}$ means the $i^{th}$ column vector of $\mathbf{U}$, and $\boldsymbol{\Sigma}_i$ is the $i^{th}$ value of the diagonal matrix $\boldsymbol{\Sigma}$.

## E EXPERIMENT DETAILS

### E.1 IMPLEMENTATION DETAILS FOR SST

**Sampling of $\mathbf{U}$ and $\mathbf{V}^{\mathrm{T}}$.** In our experiments, we employ a more exploratory approach when sampling $\mathbf{U}$ and $\mathbf{V}^{\mathrm{T}}$:

$$p(i) = \frac{1}{2}\left(\frac{1}{m} + \frac{\mathbf{\Sigma}_i}{\sum_j \mathbf{\Sigma}_j}\right) \tag{16}$$

where $p(i)$ is the possibility to sample index $i$ vector of $\mathbf{U}$ and $\mathbf{V}^{\mathrm{T}}$. This method modifies the earlier Eq. 5 by combining the multinomial distribution with a uniform distribution. This adjustment ensures that vectors associated with lower singular values still have a substantial likelihood of being sampled, preventing their probabilities from becoming excessively low and promoting a more balanced exploration across the spectral components.

**Optimizer state reset and warmup.** Before each iteration, Sparse Spectral Training (SST) resets all optimizer states for $\mathbf{U}$, $\mathbf{V}^{\mathrm{T}}$ and $\mathbf{\Sigma}$. For example, for optimizers like Adam, this involves clearing the first and second moments as well as the timestep. Consequently, a brief warmup period is essential at the beginning of each iteration to accommodate the reset states. This warmup period is typically 20 steps, guided by the exponential decay rate $\beta$ used in the Adam optimizer.

**Hyperbolic SST.** The formula of hyperbolic linear layer in (Chen et al., 2022) is:

$$\mathbf{h} = f_{\mathbf{x}}(\mathbf{M})\mathbf{x} = \begin{bmatrix} \frac{\sqrt{\|\mathbf{W}\mathbf{x}\|_2 - \frac{1}{K}}}{\mathbf{v}^{\top}\mathbf{x}}\mathbf{v}^{\top} \\ \mathbf{W} \end{bmatrix}\mathbf{x} = \begin{bmatrix} \sqrt{\|\mathbf{W}\mathbf{x}\|_2 - \frac{1}{K}}\mathbf{v}^{\top} \\ \mathbf{W}\mathbf{x} \end{bmatrix} \tag{17}$$

where $\mathbf{v} \in \mathbb{R}^{n+1}$, $\mathbf{W} \in \mathbb{R}^{m \times (n+1)}$ and $K$ is the curvature. The formula of Hyperbolic SST is:

$$h = \begin{bmatrix} \sqrt{\|\mathbf{U}\mathbf{\Sigma}\mathbf{V}^{\mathrm{T}}\mathbf{x}\|_2 - \frac{1}{K}}\mathbf{v}^{\top} \\ \mathbf{U}\mathbf{\Sigma}\mathbf{V}^{\mathrm{T}}\mathbf{x} \end{bmatrix} \tag{18}$$

### E.2  HYPERPARAMETERS OF MACHINE TRANSLATION

**IWSLT'14.** The hyperparameters can be found in Table 7. We employ the same codebase and hyperparameters as those used in HyboNet (Chen et al., 2022), which is derived from OpenNMT-py (Klein et al., 2017). For all methods, last checkpoint is utilized for evaluation. Beam search, with a beam size of 2, is employed to optimize the evaluation process. Experiments were conducted on one A100 GPU.

For SST, iteration interval ($T_3$) is set to 200. Each iteration begins with a warmup phase lasting 20 steps. The number of iterations per round ($T_2$) is determined by the formula $T_2 = d/r$, where $d$ represents the embedding dimension and $r$ denotes the rank used in SST.

**Multi30K and IWSLT'17.** The hyperparameters can be found in Table 8. Because of overfitting, model checkpoint with lowest validation loss is utilized for evaluation. A larger learning rate (0.0003) is used for low rank parameters ($\mathbf{U}$, $\mathbf{V}^{\mathrm{T}}$ and $\mathbf{\Sigma}$ for SST, $\mathbf{B}$ and $\mathbf{A}$ for LoRA and ReLoRA*). Experiments were conducted on one A100 GPU.

For SST, interation interval ($T_3$) is set to 200 for Multi30K and 400 for IWSLT'17. Each iteration begins with a warmup phase lasting 20 steps. The number of iterations per round ($T_2$) is determined by the formula $T_2 = d/r$, where $d$ represents the embedding dimension and $r$ denotes the rank used in SST.

### E.3  HYPERPARAMETERS OF NATURAL LANGUAGE GENERATION

**Hyperparameters for OPT.** The hyperparameters for OPT are detailed in Table 9. We employ a linear warmup of 2000 steps followed by a stable learning rate, without decay. A larger learning rate (0.001) is used for only low rank parameters ($\mathbf{U}$, $\mathbf{V}^{\mathrm{T}}$ and $\mathbf{\Sigma}$ for SST, $\mathbf{B}$ and $\mathbf{A}$ for LoRA and ReLoRA*). The total training tokens for each experiment is 19.7B, roughly 2 epochs of OpenWebText. Distributed training is facilitated using the Accelerate (Gugger et al., 2022) library across four A100 GPUs on a Linux server.

Table 7: **Hyperparameters on IWSLT'14** for Euclidean and hyperbolic Transformer.

| Hyper-parameter | Euclidean | Hyperbolic |
|---|---|---|
| Embedding Dimension | 64, 128, 256 | 64, 128, 256 |
| Feed-forward Dimension | 256, 512, 1024 | 256, 512, 1024 |
| Batch Size | 10240 tokens | 10240 tokens |
| Gradient Accumulation Steps | 4 | 4 |
| Training Steps | 40000 | 40000 |
| Dropout | 0.0 | 0.1 |
| Attention Dropout | 0.1 | 0.1 |
| Max Gradient Norm | - | 0.5 |
| Warmup Steps | 6000 | 6000 |
| Decay Method | noam | noam |
| Label Smoothing | 0.1 | 0.1 |
| Layer Number | 6 | 6 |
| Head Number | 4 | 4 |
| Learning Rate | 5 | 2 |
| Optimizer | Adam | rAdam |

Table 8: **Hyperparameters on Multi30K and IWSLT'17** for vanilla Transformer.

| Hyper-parameter | Multi30K | IWSLT'17 |
|---|---|---|
| Embedding Dimension | 512 | 512 |
| Feed-forward Dimension | 2048 | 2048 |
| Batch Size | 128 sentences | 128 sentences |
| Gradient Accumulation Steps | 1 | 1 |
| Training Steps | 100000 | 150000 |
| Dropout | 0.1 | 0.1 |
| Decay Method | constant | constant |
| Layer Number | 6 | 6 |
| Head Number | 8 | 8 |
| Learning Rate | 0.0001 | 0.0001 |
| Weight Decay | 1 | 0.1 |
| Optimizer | AdamW | AdamW |

For SST, interation interval ($T_3$) is set to 200. Each iteration begins with a warmup phase lasting 20 steps. The number of iterations per round ($T_2$) is determined by the formula $T_2 = d/r$, where $d$ represents the embedding dimension and $r$ denotes the rank used in SST.

Table 9: **Hyperparameters for OPT Models**

| Hyper-parameter | OPT-125M | OPT-350M | OPT-1.3B |
|---|---|---|---|
| Embedding Dimension | 768 | 512 (project to 1024) | 2048 |
| Feed-forward Dimension | 3072 | 4096 | 8192 |
| Global Batch Size | 240 | 240 | 240 |
| Sequence Length | 2048 | 2048 | 2048 |
| Training Steps | 40000 | 40000 | 40000 |
| Learning Rate | 0.0001 | 0.0001 | 0.0001 |
| Warmup Steps | 2000 | 2000 | 2000 |
| Optimizer | AdamW | AdamW | AdamW |
| Layer Number | 12 | 24 | 24 |
| Head Number | 12 | 16 | 32 |

**Hyperparameters for LLaMA.** The hyperparameters for LLaMA are detailed in Table 10. We follow the same experiment setup from (Zhao et al., 2024). We employ a linear warmup of 2000/10000

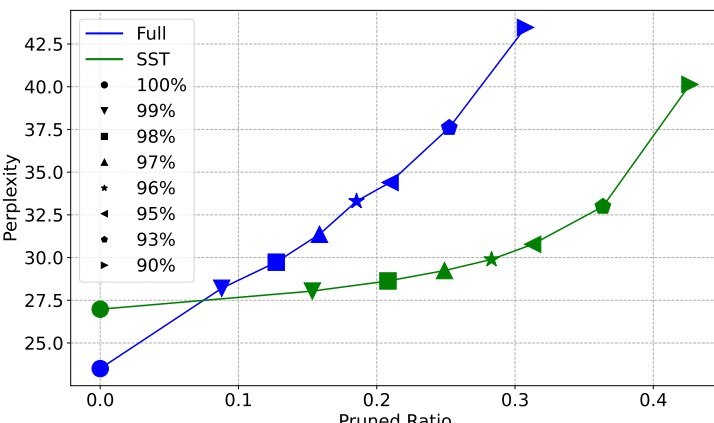

Figure 5: **Singular Value Pruning.** We conduct singular value pruning on full-rank and SST pretrained OPT-125M model. After performing singular value decomposition on weight matrices, we preserve the top $k$ singular values so that the cumulative sum of preserved singular values ranges from $[100\%, 99\%, 98\%, ..., 93\%, 90\%]$ of the original cumulative sum. The pruned ratio of singular values is plotted along the x-axis.

steps followed by a cosine decay. For LLaMA-130M, the learning rates for LoRA, ReLoRA*, and SST are selected from $\{1e\text{-}3, 3e\text{-}3\}$ based on the lowest PPL observed in Table 6. For LLaMA-1.3B, the learning rates for LoRA, ReLoRA*, and SST are fixed at 1e-3. The learning rates for full-rank training are set to 1e-3 for LLaMA-130M and 4e-4 for LLaMA-1.3B, consistent with the configuration in the ReLoRA article.

For SST, interation interval ($T_3$) is set to 200. Each iteration begins with a warmup phase lasting 20 steps. The number of iterations per round ($T_2$) is determined by the formula $T_2 = d/r$, where $d$ represents the embedding dimension and $r$ denotes the rank used in SST.

Table 10: **Hyperparameters for LLaMA Models**

| Hyper-parameter | LLaMA-130M | LLaMA-1.3B |
|---|---|---|
| Embedding Dimension | 768 | 2048 |
| Feed-forward Dimension | 2048 | 5461 |
| Global Batch Size | 512 | 512 |
| Sequence Length | 256 | 256 |
| Training Steps | 20000 | 100000 |
| Learning Rate | 0.001 | 0.0004 |
| Warmup Steps | 2000 | 10000 |
| Optimizer | Adam | Adam |
| Layer Number | 12 | 24 |
| Head Number | 12 | 32 |

### E.4 Hyperparameters of Hyperbolic Graph Neural Networks

We use HyboNet (Chen et al., 2022) as full-rank model, with same hyperparameters as those used in HyboNet. Experiments were conducted on one A100 GPU.

For SST, interation interval ($T_3$) is set to 100. Each iteration begins with a warmup phase lasting 100 steps. The number of iterations per round ($T_2$) is determined by the formula $T_2 = d/r$, where $d$ represents the embedding dimension and $r$ denotes the rank used in SST.

We set dropout rate to 0.5 for the LoRA and SST methods during the node classification task on the Cora dataset. This is the only one deviation from the HyboNet configuration.

## F  Singular Value Pruning

We further conduct an analysis study of the potential for using SST model for further compression. The results, as shown in Figure 5, indicate that the SST model retains lower perplexity across a wider range of pruning ratios compared to the full-rank model. This suggests that the SST method effectively concentrates the informational content of the weights into fewer singular values, making it more suitable for further compression.

This enhanced performance underscores the potential of SST in maintaining essential model characteristics even under significant compression, making it a promising approach for developing lightweight yet powerful language models for inference.

## G  Evaluating SST and GaLore: Complementary Approaches to Memory Efficiency

Table 11: **The BLEU score on IWSLT'14 for Euclidean Transformer, compared with GaLore.** Values highlighted in bold represent the highest performance among the low rank methods, while those marked with an "*" denote performance that exceeds that of the full-rank variants.

| Dimension | r | Full | GaLore | SST |
|---|---|---|---|---|
| 64 | 8 | 24.27 | 18.08 | **22.28** |
|  | 4 |  | 14.07 | **20.27** |
| 128 | 16 | 25.79 | 23.43 | **25.12** |
|  | 8 |  | 19.71 | **24.19** |
|  | 4 |  | 16.01 | **22.80** |
| 256 | 32 | 23.92 | **24.01*** | 23.97* |
|  | 16 |  | 22.82 | **23.42** |
|  | 8 |  | 20.12 | **22.65** |
|  | 4 |  | 15.94 | **21.39** |

Recently, a new approach named Gradient Low-Rank Projection (GaLore) (Zhao et al., 2024) has been proposed to address the memory challenges associated with pre-training large language models. GaLore, by implementing a memory-efficient gradient projection method.

Using the released code of GaLore[1], we conducted comparative experiments on the IWSLT'14 dataset with Transformer models, employing the same configurations as other low-rank methods. We set the scale factor $\alpha = 1$ in these experiments because $\alpha = 0.25$, which is used in the article, performs much worse than $\alpha = 1$. As illustrated in Table 11, SST method consistently outperformed GaLore across various model dimensions and ranks, except for $d = 256, r = 32$.

In addition, we evaluated validation perplexity on the OpenWebText dataset with OPT-125M and OPT-350M models. We tested GaLore with scale factor $\alpha = 0.25$ (used in GaLore article) and $\alpha = 1$. As shown in Table 12, SST outperformed GaLore at both settings of $\alpha$ on OPT-125M. Since $\alpha = 1$ had better results than $\alpha = 0.25$ on OPT-125M, we used $\alpha = 1$ for training GaLore on OPT-350M. Initially, GaLore trained normally on OPT-350M, but around step 6127, the training loss suddenly increased from approximately 4 to 7 within a few steps, resulting in a very high final perplexity for the GaLore OPT-350M, as shown in Table 12. Training of GaLore on OPT-1.3B is still ongoing, and we will update the results as soon as they are available. Zero-shot evaluations comparing SST with GaLore are presented in Table 13, which also demonstrate SST's superior performance.

Here, we discuss our guess on why SST may have an advantage over GaLore on low-rank settings. GaLore utilizes a projection matrix $P_t \in \mathbb{R}^{m \times r}$ derived from the singular value decomposition (SVD) of a single step's gradient. Only one step's gradient may introduce noise due to data sampling variability. Conversely, SST employs $\mathbf{U}$ and $\mathbf{V}^{\mathrm{T}}$ as projection matrices, which are initialized and reinitialized with the SVD of $\mathbf{W}$. $\mathbf{W}$ could be seemed as the momentum of gradient of $\mathbf{W}$, less noisy

---

[1] https://github.com/jiaweizzhao/GaLore

than one step's gradient. Furthermore, SST updates all $\Sigma$ values, regardless of $r$, making it more robust as $r$ decreases.

Table 12: **Validation perplexity, compared with GaLore** on OpenWebText dataset with OPT-125M and OPT-350M, along with the number of trainable parameters of each method. $r = 64$. Values highlighted in bold represent the highest performance among the low rank methods.

|  | **Full** | **GaLore** $\alpha = 1$ | **GaLore** $\alpha = 0.25$ | **SST** |
|---|---|---|---|---|
| **OPT-125M** | 23.50 (125.2M) | 32.17 (45.6M) | 37.08 (45.6M) | **26.98** (51.0M) |
| **OPT-350M** | 21.78 (331.2M) | 1994 (43.4M) | - | **27.72** (57.7M) |

Table 13: **Zero-shot evaluations, compared with GaLore** with same tasks as Table 4. Mean scores in bold represent superior performance among the low-rank methods. Win percentage (including ties) for each low-rank method is compared to the full-rank training.

|  | **OPT-125M** | | | | **OPT-350M** | | |
|---|---|---|---|---|---|---|---|
|  | Full | GaLore $\alpha = 1$ | GaLore $\alpha = 0.25$ | SST | Full | GaLore $\alpha = 1$ | SST |
| ARC (Challenge) | 21.2 | 21.2 | 20.4 | 21.3 | 22.0 | 25.7 | 21.1 |
| ARC (Easy) | 35.8 | 33.7 | 32.8 | 34.3 | 35.9 | 25.7 | 35.7 |
| BoolQ | 59.5 | 61.8 | 62.2 | 62.0 | 53.6 | 37.8 | 57.7 |
| CB | 51.8 | 37.5 | 35.7 | 48.2 | 44.6 | 41.1 | 41.1 |
| COPA | 67.0 | 64.0 | 58.0 | 66.0 | 69.0 | 52.0 | 60.0 |
| HellaSwag | 27.7 | 27.0 | 26.6 | 26.9 | 28.4 | 26.2 | 27.5 |
| MultiRC | 55.4 | 57.2 | 54.8 | 57.2 | 52.0 | 42.8 | 57.0 |
| OpenBookQA | 24.6 | 23.6 | 24.6 | 26.2 | 26.4 | 27.8 | 25.2 |
| PIQA | 58.7 | 57.1 | 56.4 | 58.3 | 59.2 | 50.5 | 59.0 |
| ReCoRD | 16.7 | 15.0 | 16.4 | 18.5 | 19.4 | 17.5 | 23.2 |
| RTE | 50.5 | 51.6 | 56.0 | 53.4 | 52.0 | 52.7 | 50.2 |
| StoryCloze | 55.8 | 53.5 | 52.8 | 54.5 | 57.2 | 49.7 | 54.6 |
| WIC | 49.8 | 50.0 | 50.0 | 50.0 | 50.5 | 50.0 | 50.2 |
| Winograd | 52.0 | 50.9 | 52.4 | 50.6 | 55.0 | 50.2 | 51.3 |
| Winogrande | 49.1 | 51.7 | 48.4 | 50.1 | 50.7 | 49.4 | 52.0 |
| WSC | 36.5 | 36.5 | 36.5 | 36.5 | 36.5 | 63.5 | 36.5 |
| Mean | 44.5 | 43.3 | 42.8 | **44.6** | 44.5 | 41.4 | **43.9** |
| Win Percentage | - | 43.8 | 37.5 | 56.3 | - | 25.0 | 31.3 |

# H ABLATION STUDY

**Impact of $\Sigma$ updates.** We conduct an ablation study to evaluate the impact of various components and configurations within SST on the IWSLT'14 using a Euclidean Transformer with a dimension of 128 and rank $r$ of 4. The results of this study are summarized in Table 14, which highlights the contributions of specific elements to the overall performance measured in BLEU score.

One variation tested involves changing the update mechanism for $\Sigma$. Instead of updating all $\Sigma$, only sampled $\Sigma$ are updated, same as update for $\mathbf{U}$ and $\mathbf{V}^\mathrm{T}$. This modification results in a lower BLEU score of 22.40, indicating that full updates of $\Sigma$ contribute positively to the model's performance.

**Initialization method.** We experiment with a configuration similar to the ReLoRA*, where $\mathbf{h} = (\mathbf{W} + \mathbf{U\Sigma V}^\mathrm{T})\mathbf{x}$, with $\mathbf{U}$ and $\mathbf{V}^\mathrm{T}$ randomly initialized and $\Sigma$ initialized to zero. After each round, $\mathbf{U}$, $\mathbf{V}^\mathrm{T}$ and $\Sigma$ are reinitialized. This setup significantly reduces the BLEU score to 16.03, which is similar to the performance of LoRA (16.37) and ReLoRA* (18.00). This demonstrates that the most important feature of SST is that instead of randomly initialized, SST uses SVD of $\mathbf{W}$ as the initialization of $\mathbf{U}$ and $\mathbf{V}^\mathrm{T}$, which is aligned with our analysis in section 4.3.

**Impact of iteration interval** ($T_3$)**.** We also conducted additional experiments to study the impact of varying iteration interval $T_3$ (sampling period). All methods were trained on a vanilla Transformer

Table 14: **Ablation Study** on IWSLT'14 dataset with Euclidean Transformer. Dimension is 128 and $r$ is 4.

| | BLEU |
|---|---|
| LoRA | 16.37 |
| ReLoRA* | 18.00 |
| SST - Instead of update all $\mathbf{\Sigma}$, only update sampled $\mathbf{\Sigma}$ | 22.40 |
| SST - Use formula similar as ReLoRA*: $\mathbf{h} = (\mathbf{W} + \mathbf{U\Sigma V}^{\mathrm{T}})\mathbf{x}$. ($\mathbf{U}$ and $\mathbf{V}^{\mathrm{T}}$ random initialized, and $\mathbf{\Sigma}$ zero initialized) | 16.03 |
| SST | 22.80 |

model with a hidden dimension of 64 and $r = 8$ on the IWSLT'14 dataset. In the original setup (Table 1), $T_3$ was set to 200 steps per iteration.

Table 15: **Impact of iteration interval** ($T_3$) on BLEU scores for IWSLT'14.

| Steps per Iteration $T_3$ | 800 | 400 | 200 | 100 | 50 | 25 | 10 |
|---|---|---|---|---|---|---|---|
| BLEU Score | 21.85 | 23.64 | 22.47 | 22.49 | 22.60 | 22.46 | 22.25 |

As shown in Table 15, both excessively large and small values of $T_3$ result in decreased performance. A large $T_3$ may cause SST degrade to LoRA, while a small $T_3$ leads to frequent resets of the optimizer's momentum, thereby affecting convergence.

**Impact of Number of Iterations.** We conducted an additional experiment on the IWSLT'14 dataset using a vanilla Transformer to evaluate the impact of the number of iterations per round, with a model dimension of 64 and $r = 8$. The results are summarized in Table 16:

Table 16: **Impact of number of iterations** per round on BLEU scores for IWSLT'14.

| Number of Iterations per Round | 1 | 2 | 4 | 8 | 16 | 32 |
|---|---|---|---|---|---|---|
| BLEU Score | 22.28 | 22.21 | 22.24 | 22.28 | 22.30 | 22.37 |

The results indicate that different numbers of iterations yield comparable performance. In our experiments, this hyperparameter was not tuned; instead, we fixed it to $d/r$.

**Sampling Mechanisms.** To evaluate the impact of different sampling mechanisms on the performance of SST, we conducted additional experiments using a vanilla Transformer with a model dimension of 64 and $r = 8$ on the IWSLT'14 dataset. The evaluation metric is BLEU, where higher scores indicate better performance. Table 17 summarizes the results:

Descriptions of Sampling Mechanisms:

- **MULTINOMIAL**: The multinomial random sampling method used in SST.
- **UNIFORM**: Uniform random sampling.
- **SEQUENTIAL**: Iterating through all singular vectors without repetition.
- **TOP_R**: Selecting the top-$r$ singular vectors with the largest singular values.

We also considered a Binomial sampling mechanism; however, it could not guarantee that the number of selected singular vectors would remain consistent with the specified rank, making it unsuitable for direct comparison.

The results indicate that **TOP_R** performs the worst, as its search space collapses into a restricted low-rank subspace. In contrast, as long as all singular vectors are visited, the other methods deliver comparable performance. Among these, **MULTINOMIAL** demonstrates a slight advantage.

Table 17: BLEU scores for different **sampling mechanisms** on IWSLT'14. Bold indicates the highest performance.

| Sampling Mechanism | MULTINOMIAL | UNIFORM | SEQUENTIAL | TOP_R |
|---|---|---|---|---|
| BLEU | **22.28** | 22.01 | 22.13 | 18.28 |

**Impact of Rank.** For all low-rank methods, including LoRA, ReLoRA*, and SST, rank is more of a constraint determined by available resources rather than a hyperparameter to be extensively tuned. Higher ranks generally lead to better performance but at the cost of increased memory consumption. To ensure fairness, the same rank values were used for LoRA, ReLoRA*, and SST in all experiments, as these methods have a similar number of trainable parameters under the same rank.

Additionally, we conducted an experiment on the IWSLT'14 dataset using a vanilla Transformer with a model dimension of 128 to analyze the impact of rank on different methods. The results are presented in Table 18:

Table 18: **Impact of rank** on BLEU scores for IWSLT'14. Dimension is 128.

| Rank ($r$) | 1 | 2 | 4 | 8 | 16 | 32 | 64 |
|---|---|---|---|---|---|---|---|
| **LoRA** | 12.44 | 14.16 | 16.37 | 20.56 | 23.30 | 25.12 | 26.11 |
| **ReLoRA** | 14.53 | 15.39 | 18.00 | 20.61 | 22.92 | 24.15 | 25.25 |
| **SST** | 17.49 | 20.69 | 22.80 | 24.19 | 25.12 | 26.08 | 26.15 |

The evaluation metric is BLEU, where higher scores indicate better performance. The BLEU score for full-rank training is 25.79. The results demonstrate that as the rank increases, the performance of all methods improves. Notably, SST consistently outperforms other low-rank methods, especially at smaller ranks, highlighting its robustness under resource-constrained settings.

**Impact of Training Steps.** To investigate whether additional training steps benefit SST, we conducted an experiment on the IWSLT'14 dataset using a vanilla Transformer with a model dimension of 64 and $r = 4$. Table 19 presents the BLEU scores for full-rank training and SST under different training steps (evaluated on the model at the last step):

Table 19: BLEU scores under **different training steps.** The default training step in Table 1 is 40,000.

| Steps | 20,000 | 40,000 | 80,000 | 160,000 | 320,000 | 640,000 |
|---|---|---|---|---|---|---|
| **Full** | 22.95 | 24.27 | 24.85 | 24.72 | 24.71 | 25.05 |
| **SST** | 17.23 | 20.27 | 21.91 | 22.86 | 23.32 | 23.92 |

The results demonstrate that as the number of training steps increases, the gap between full-rank training and SST narrows. Even with $r = 4$, SST approaches the performance of full-rank training at 640,000 steps. These findings confirm that while SST may require more steps to converge at lower ranks, it remains competitive with full-rank training given sufficient steps.

## I MEMORY CONSUMPTION AND TRAINING TIME

**Memory consumption.** As shown in Table 20, the memory consumption of SST is comparable to LoRA and much smaller than full-rank models. SST has a similar number of trainable parameters (about 0.2% higher) as LoRA (as stated in Table 3), but more frozen parameters (about 45% higher) than LoRA. However, this can be mitigated if we use low precision for the frozen parameters, as in (Dettmers et al., 2024).

Table 21 shows that the memory consumption of SVD decomposition for the largest weight in each model is about 3%, which is small compared with the whole model.

Table 20: **GPU memory consumption** on different sizes of OPT models, including optimizer state and gradient. Model weight uses float32. AdamW optimizer state uses float32 (same data type as used in OPT experiments in Table 3).

|          | Full        | LoRA/ReLoRA* | SST        |
|----------|-------------|--------------|------------|
| OPT-125M | 1956.05 MB  | 1118.56 MB   | 1254.41 MB |
| OPT-350M | 5070.41 MB  | 2046.41 MB   | 2573.77 MB |
| OPT-1.3B | 20093.22 MB | 7133.24 MB   | 9345.72 MB |

Table 21: **GPU memory consumption of SVD decomposition in SST.**

| Model    | Largest Weight Shape | Peak GPU Memory Consumption |
|----------|----------------------|-----------------------------|
| OPT-125M | $768 \times 3072$    | 41.25 MB (3.3%)             |
| OPT-350M | $1024 \times 4096$   | 72.00 MB (2.8%)             |
| OPT-1.3B | $2048 \times 8192$   | 288.01 MB (3.1%)            |

**Training time.** Table 22 shows that the time spent on SVD in SST is very low, about 0.5%-0.8% compared with the whole training time. SST has comparable training time as LoRA and full-rank model. The increasement of training time of SST is mainly due to SST's linear function, $\mathbf{h} = \mathbf{U\Sigma V}^{\mathrm{T}}\mathbf{x}$, which is slower than original $\mathbf{h} = \mathbf{Wx}$. However, during inference, replacing $\mathbf{U\Sigma V}^{\mathrm{T}}$ with a single matrix $\mathbf{W}$ could obtain same computation efficiency as full-rank models. ReLoRA* has comparable computation time as LoRA.

Table 22: **Overall training time** on different sizes of OPT models with 19.7 billion training tokens, using 4 A100 GPU. "Time of SVD in SST" is the overall time of singular value decomposition within SST.

| Model    | Full   | LoRA   | SST    | Time of SVD in SST |
|----------|--------|--------|--------|--------------------|
| OPT-125M | 62.5h  | 64.4h  | 65.0h  | 0.3h (0.5%)        |
| OPT-350M | 135.8h | 153.3h | 170.0h | 0.8h (0.5%)        |
| OPT-1.3B | 303.4h | 324.8h | 387.2h | 3.0h (0.8%)        |

**Performance with Fewer Steps.** Despite requiring slightly more time per step, SST achieves superior performance with fewer training steps compared to other low-rank methods. The choice of 20% fewer steps for SST corresponds to the maximum additional training time incurred by SST compared to other low-rank methods, as shown in Table 22. Table 23 compares the perplexity (PPL) of SST trained with 20% fewer steps to that of other methods trained with full steps.

These results demonstrate that SST maintains significantly lower perplexity even with fewer training steps, highlighting its efficiency. SST effectively balances its computational overhead while achieving superior performance compared to other low-rank methods. This makes SST a compelling choice for high-quality pretraining.

## J EXPERIMENT ON IMAGE CLASSIFICATION

We conduct additional experiments on image classification tasks using MLP-based models. In this section, we provide a comparison of full-rank training, LoRA, ReLoRA*, and SST on three datasets: MNIST (Lecun et al., 1998), EMNIST (Cohen et al., 2017), and Fashion_MNIST (Xiao et al., 2017).

The architecture of the MLP is $784 - 512 - 512 - 512 - $#class. Each method is trained for a total of 100 epochs. Learning rate is set to 0.01 for all methods.

Table 23: **Validation perplexity with SST trained 20% fewer steps** compared to full steps for other methods.

| Model | Full | LoRA | ReLoRA* | SST (20% fewer steps) |
|---|---|---|---|---|
| OPT-125M | 23.50 | 34.23 | 35.80 | **28.03** |
| OPT-350M | 21.78 | 34.26 | 39.21 | **29.42** |
| OPT-1.3B | 15.10 | 1716 | 29.52 | **22.98** |
| LLaMA-130M | 20.04 | 29.71 | 31.33 | **24.74** |
| LLaMA-1.3B | 14.54 | 16.50 | 17.32 | **15.65** |

We use a rank of 16 for all low-rank methods, which corresponds to 1/32 of the full-rank dimension. For ReLoRA* and SST, one epoch per iteration is used. The results are averaged over three random seeds, and all datasets were evaluated based on test accuracy.

Table 24: **Image classification tasks** test accuracy.

| Dataset | Full | LoRA | ReLoRA* | SST |
|---|---|---|---|---|
| MNIST | 98.63 ± 0.04 | 97.69 ± 0.10 | 97.72 ± 0.05 | 98.33 ± 0.04 |
| EMNIST | 85.32 ± 0.24 | 79.45 ± 0.26 | 84.12 ± 0.12 | 84.96 ± 0.11 |
| Fashion_MNIST | 90.44 ± 0.06 | 88.30 ± 0.01 | 89.08 ± 0.16 | 89.22 ± 0.06 |

As shown in Table 24, SST outperforms both LoRA and ReLoRA* across all three datasets. SST reduces performance gap between low-rank method and full-rank training by **49%** in average.

## K   MEMORY EFFICIENCY ANALYSIS

To better understand the memory efficiency of SST compared to baseline methods, we provide a detailed joint analysis of GPU memory consumption and performance trade-offs.

**Memory and Performance Trade-Off.**   SST's GPU memory consumption is comparable to ReLoRA*, while achieving significant improvements in perplexity (PPL). A comparison of memory reduction and PPL increase is provided in our analysis (Figure 6).

We define the following metrics for clarity:

$$\text{Memory Reduction (\%)} = \frac{\text{Full memory} - \text{Low rank memory}}{\text{Full memory}} \times 100$$

$$\text{PPL Increase (\%)} = \frac{\text{Low rank PPL} - \text{Full PPL}}{\text{Full PPL}} \times 100$$

To provide a more intuitive understanding of SST's memory efficiency, we introduce a new metric called the **efficiency ratio**, defined as:

$$\text{Efficiency Ratio} = \frac{\text{Memory Reduction (\%)}}{\text{PPL Increase (\%)}}$$

This efficiency ratio quantifies how much memory can be reduced at the cost of a 1% increase in PPL. A higher efficiency ratio indicates a more memory-efficient method.

**Results.**   SST achieves a significantly higher efficiency ratio than ReLoRA* across various pretraining tasks. Figure 7 shows the efficiency ratio improvements of SST compared to ReLoRA*:

- **167.4%** (OpenWebText, LLaMA-130M)
- **99.7%** (C4, LLaMA-130M)

- **196.1%** (OpenWebText, OPT-125M)
- **142.3%** (OpenWebText, OPT-350M)
- **65.9%** (OpenWebText, OPT-1.3B)
- **4434.3%** (OpenWebText, LLaMA-1.3B)

**Conclusion.** These results demonstrate that SST achieves a substantially better trade-off between memory reduction and PPL increase compared to ReLoRA*. This highlights SST's effectiveness in optimizing memory efficiency while maintaining strong model performance, making it a practical choice for resource-constrained pretraining tasks.

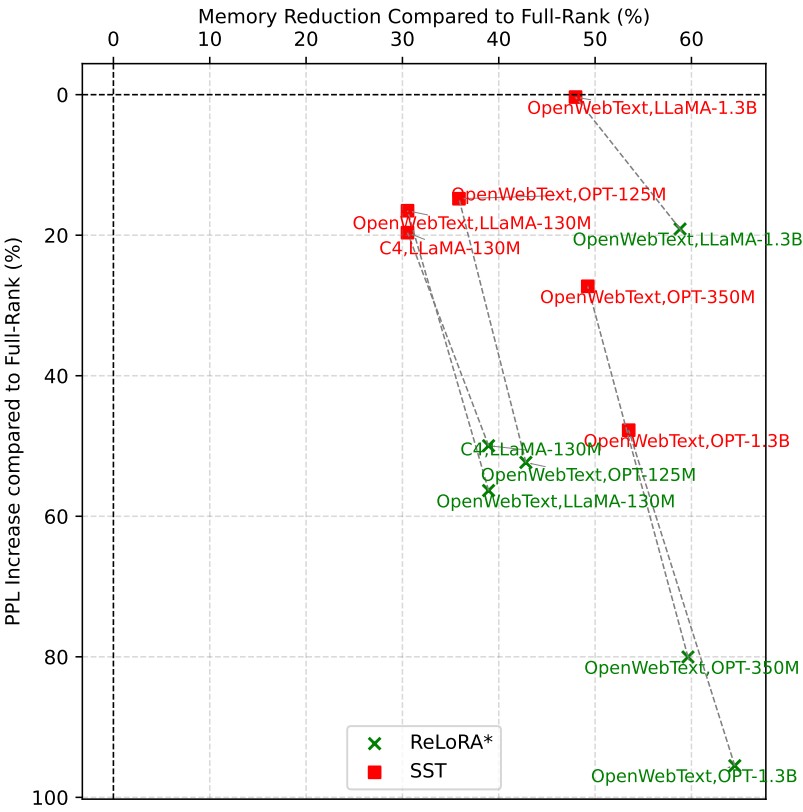

Figure 6: **Memory reduction vs. PPL increase.** Comparison of SST and ReLoRA* on multiple datasets and models.

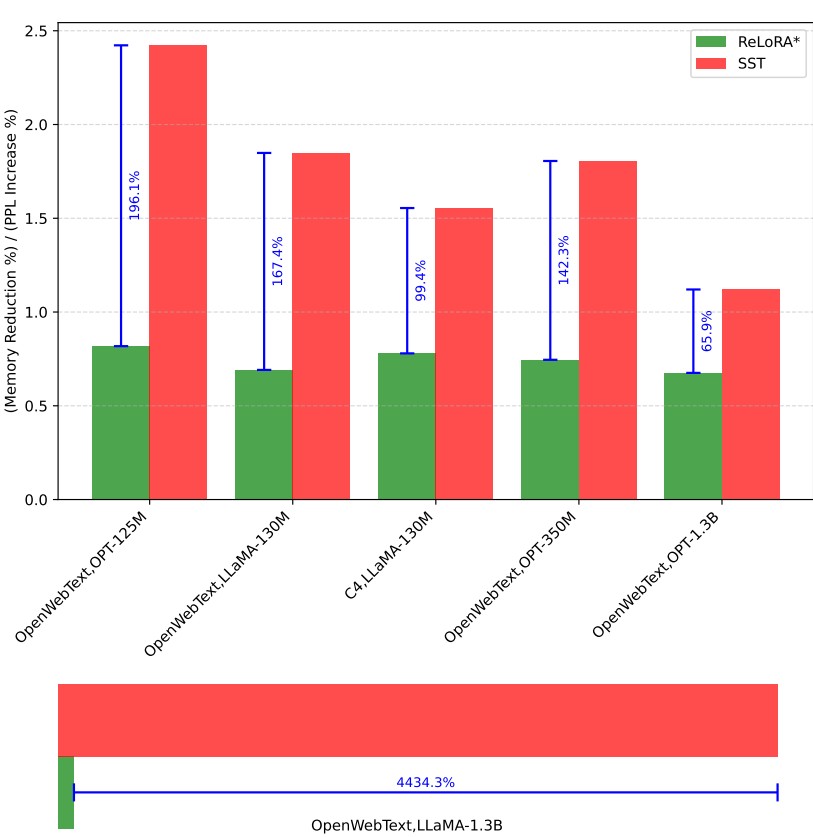

Figure 7: **Efficiency Ratio Improvements.** SST achieves significantly higher efficiency ratios compared to ReLoRA* across various tasks and model sizes. The LLaMA-1.3B result is included at the bottom of the plot due to its large value.

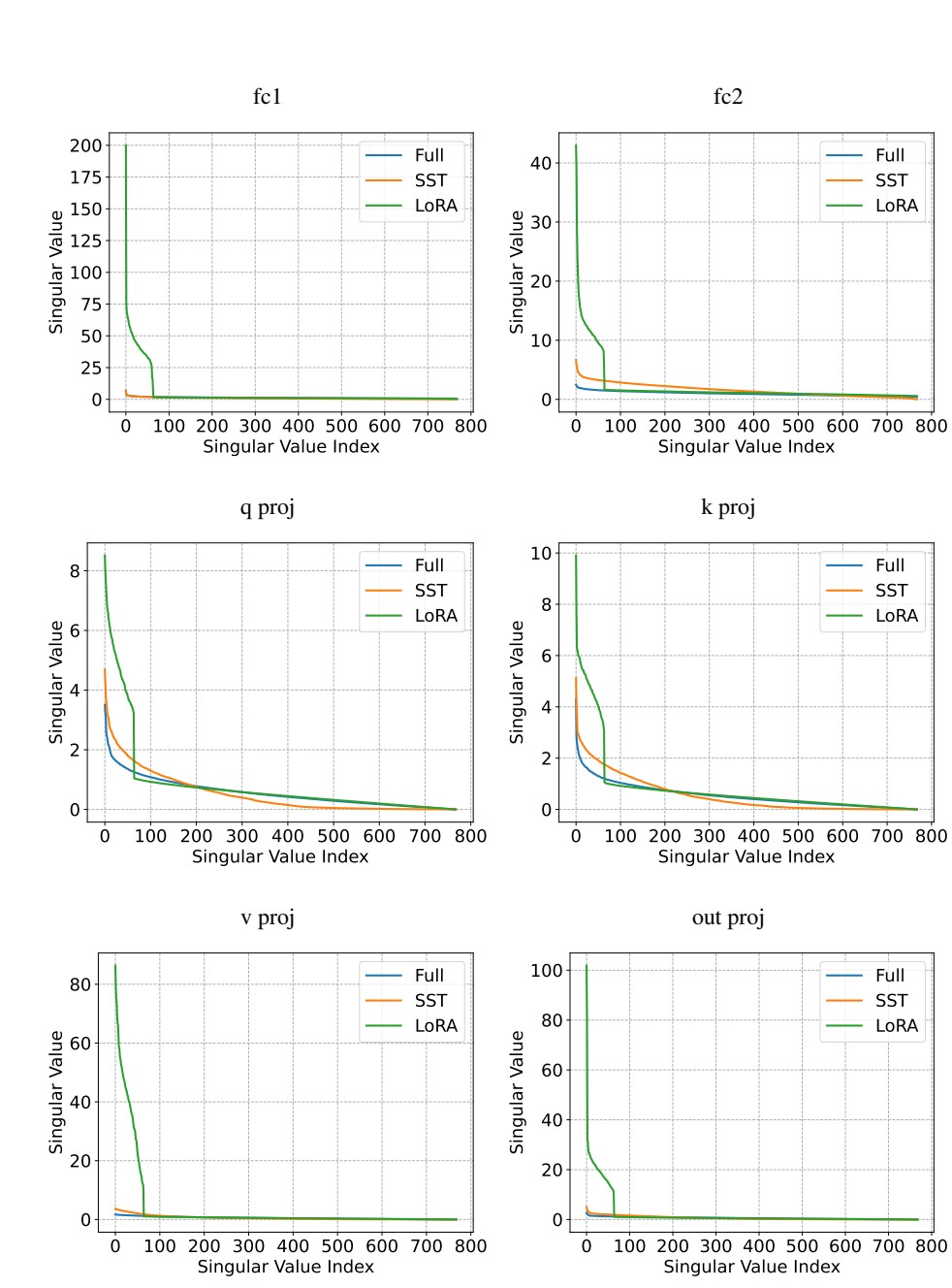

Figure 8: **Singular Value Distribution.** This visualization depicts the distribution of singular values for the OPT-125M model with full-rank, LoRA, and SST, with $r = 64$). The x-axis represents the index of singular values, sorted from largest to smallest, while the y-axis shows the magnitude of each value. It highlights how LoRA predominantly captures and overestimates the top-$r$ singular values, in contrast to SST, which shows a much similar distribution as full-rank training.

