# OpenReview forum: "Sparse Spectral Training and Inference on Euclidean and Hyperbolic Neural Networks"
_ICLR.cc/2025/Conference — Submitted to ICLR 2025_

### Official Review · Reviewer_TBSo · 2024-10-30

**Soundness:** 3
**Presentation:** 2
**Contribution:** 3
**Rating:** 6
**Confidence:** 3

**Summary:**

By selectively updating singular values ​​and vectors in the weight matrix, this paper proposes Sparse Spectral Training (SST), a memory-efficient technique for training large-scale neural networks. SST reduces memory usage while maintaining comparable performance to full-rank training by using SVD for startup and periodic reinitialization. SST outperforms current low-rank techniques such as LoRA and ReLoRA in machine translation and natural language processing, and shows competitive results when tested on a variety of neural network architectures and tasks

**Strengths:**

1. By focusing on selective updates, SST provides an efficient memory management technique that significantly reduces GPU memory usage during training.

2. The proposed method reduces the perplexity and BLEU score gaps on benchmarks, showing competitive or superior performance on a range of neural network architectures and tasks, often approaching the effect of full-rank training.

3. Regular use of SVD helps SST alleviate the saddle point problem of ReLoRA, bringing its training dynamics closer to full-rank training.

**Weaknesses:**

1. Frequent SVD reinitialization may affect scalability for very large models and incur significant computational overhead.

2. As with other low-rank techniques, stability issues can still occur in very low-rank models even with the sampling mechanism of SST.

3. Users who are not familiar with these techniques may find it challenging to get started, as memory efficiency optimizations (such as dividing active and frozen parts) and selective update methods may make the implementation more complex.

**Questions:**

1. How does periodic SVD reinitialization affect scalability as model size increases, especially in larger language models or high-dimensional settings?
2. How sensitive is the polynomial sampling strategy to the selection of singular values? Can the stability and efficiency of specific application scenarios be further improved by adopting different sampling methods?
3. Does periodic SVD reinitialization affect the convergence speed of models that are more constrained by low rank? In this case, is it possible that SST requires more training steps to achieve convergence compared to full-rank training?
4. Can the authors provide more practical guidance to help tune SST's hyperparameters (such as SVD reinitialization frequency) to achieve the best balance of memory efficiency and performance in different tasks?

---

> ### Author Response · Authors · 2024-11-22
> **Reply to the Reviewer TBSo (1/3)**
>
> Thank you for your detailed review and valuable insights. We are glad to address your concerns and provide clarifications.
>
> **Weakness 1:** Frequent SVD reinitialization may affect scalability for very large models and incur significant computational overhead.
>
> **Reply:** Thank you for your feedback. Both the memory consumption and training time required for SVD reinitialization are negligible compared to the overall pretraining process. As reported in Table 15 in the appendix, the peak GPU memory consumption for SVD accounts for only **3\%** of the total memory usage. This is because the SVD operation is applied to each weight matrix individually. Similarly, Table 16 in the appendix shows that the time spent on SVD reinitialization is less than **1\%** of the total training time, as reinitialization occurs at intervals of several thousand steps.
>
> We would also like to emphasize the importance of using SVD and re-SVD for effective pretraining. In Table 12 of our ablation study, we compare the performance of SST with random initialization for trainable low-rank matrices (similar to ReLoRA*) and observe a significant drop in BLEU score from 22.80 (original SST) to 16.03. This demonstrates that SST's approach of sampling singular vectors instead of using random initialization is critical for achieving strong pretraining performance. The comparison below further highlights the advantages of SST:
>
> |                        | LoRA   | ReLoRA* | SST - random initialization | SST       |
> |------------------------|--------|---------|-----------------------------|-----------|
> | **BLEU**               | 16.37  | 18.00   | 16.03                       | **22.80** |
>
> These results illustrate that while SVD introduces a small computational cost, it is essential for capturing meaningful structure during pretraining. Random initialization, although faster, fails to provide the structured starting point necessary for effective optimization, leading to significantly worse performance. We hope this clarification addresses your concerns regarding scalability and computational overhead.
>
>
> **Weakness 2:** As with other low-rank techniques, stability issues can still occur in very low-rank models even with the sampling mechanism of SST.
>
> **Reply:** Sorry, we didn't fully understand what is meant by stability issues still occurring in SST. In our article, instability refers to a sudden increase in training loss. For example, in Figure 4, we show that LoRA's loss on OPT-1.3B suddenly increases at around 0.4 × 10^3 effective steps, which also happens occasionally with GaLore. Similarly, in Table 1, we demonstrate that LoRA and ReLoRA* can result in NaN losses for certain configurations of the Hyperbolic Transformer. However, we did not observe any similar sudden increases in training loss during any of SST's pretraining experiments.
>
> If the reviewer refers to stability as robustness, we conducted an additional experiment on IWSLT'14 using a vanilla Transformer with a model dimension of 128 to evaluate the performance of different methods under varying ranks. The results are as follows:
>
>
> | r    | 1     | 2     | 4     | 8     | 16    | 32    | 64    |
> |------|-------|-------|-------|-------|-------|-------|-------|
> | LoRA | 12.44 | 14.16 | 16.37 | 20.56 | 23.30 | 25.12 | 26.11 |
> | ReLoRA | 14.53 | 15.39 | 18.00 | 20.61 | 22.92 | 24.15 | 25.25 |
> | SST  | 17.49 | 20.69 | 22.80 | 24.19 | 25.12 | 26.08 | 26.15 |
>
> The evaluation metric is BLEU, where higher scores indicate better performance. The results show that SST consistently outperforms LoRA and ReLoRA across all ranks, particularly at smaller ranks. This demonstrates SST's robustness to changes in rank, making it more stable than other low-rank methods when handling very low-rank constraints.
>
>
>
> **Weakness 3:** Users who are not familiar with these techniques may find it challenging to get started, as memory efficiency optimizations (such as dividing active and frozen parts) and selective update methods may make the implementation more complex.
>
> **Reply:** Thank you for your feedback. As demonstrated in our code (https://anonymous.4open.science/r/sparse_spectral_training-6A2C/), all the complexity of dividing active and frozen parts, as well as implementing selective update methods, is encapsulated within an `SVDLinear` class. This class seamlessly replaces the `nn.Linear` layer in the original model, making it straightforward to use SST. Additionally, we plan to integrate SST into the PEFT library (https://github.com/huggingface/peft) after acceptance. This integration will further simplify the process, requiring only a single line of code: `model = get_peft_model(model, peft_config)`.
>
>
>
> **Question 1:** How does periodic SVD reinitialization affect scalability as model size increases, especially in larger language models or high-dimensional settings?
>
> **Reply:** Thank you for this question. Please see the reply that we reported above to address the Weakness 1.

---

> ### Author Response · Authors · 2024-11-22
> **Reply to the Reviewer TBSo (2/3)**
>
> **Question 2:** How sensitive is the polynomial sampling strategy to the selection of singular values? Can the stability and efficiency of specific application scenarios be further improved by adopting different sampling methods?
>
> **Reply:** Thank you for this insightful question. We use a multinomial sampling strategy rather than a polynomial one in SST. To explore the sensitivity of SST to different sampling methods, we conducted additional experiments comparing various sampling mechanisms:
>
> | Sampling Method | MULTINOMIAL | UNIFORM | SEQUENTIAL | TOP_R |
> |------------------|-------------|---------|------------|-------|
> | **BLEU**         | **22.28**   | 22.01   | 22.13      | 18.28 |
>
> These experiments were performed with SST on a vanilla Transformer with a dimension of 64 and $r = 8$, using the IWSLT'14 dataset. BLEU was used as the evaluation metric, where higher scores indicate better performance.
>
> - **MULTINOMIAL**: The multinomial random sampling method implemented in SST.
> - **UNIFORM**: Sampling singular vectors at uniform random.
> - **SEQUENTIAL**: Iterating through singular vectors without repetition.
> - **TOP_R**: Always selecting the top-$r$ singular vectors with the largest singular values.
>
> The results suggest that the TOP_R strategy performs the worst, as it restricts the search space to a low-rank subspace, limiting exploration. Other methods, such as UNIFORM, SEQUENTIAL, and MULTINOMIAL, show comparable performance as long as all singular vectors are considered. Among these, MULTINOMIAL shows a little advantage, which supports its use in SST for balancing exploration and exploitation.
>
> We appreciate the reviewer for raising this question, and we have included these findings in the revised manuscript to provide additional insights.
>
>
> **Question 3:** Does periodic SVD reinitialization affect the convergence speed of models that are more constrained by low rank? In this case, is it possible that SST requires more training steps to achieve convergence compared to full-rank training?
>
> **Reply:** With a lower rank, SST may indeed take longer to achieve comparable performance to full-rank training. Theoretically, SST can only match full-rank training once all singular vectors have been sampled at least once. This process naturally takes longer with fewer singular vectors sampled per step, i.e., when the rank is lower.
>
> To investigate whether additional training steps benefit SST, we conducted an experiment on IWSLT'14 using a vanilla Transformer with a model dimension of 64 and $r = 4$. The BLEU scores for full-rank training and SST at different training steps are shown below (evaluated on the model at the last step):
>
> | Steps   | 20,000 | 40,000 | 80,000 | 160,000 | 320,000 | 640,000 |
> |---------|--------|--------|--------|---------|---------|---------|
> | Full    | 22.95  | 24.27  | 24.85  | 24.72   | 24.71   | 25.05   |
> | SST     | 17.23  | 20.27  | 21.91  | 22.86   | 23.32   | 23.92   |
>
> The default training step in Table 1 of the manuscript is 40,000. The results demonstrate that as the number of training steps increases, the gap between full-rank training and SST narrows. Even with $r = 4$, SST becomes comparable to full-rank training at 640,000 steps.
>
> Thank you for raising this important point. We have added this as an additional ablation study in the revised manuscript.

---

> ### Author Response · Authors · 2024-11-22
> **Reply to the Reviewer TBSo (3/3)**
>
> **Question 4:** Can the authors provide more practical guidance to help tune SST's hyperparameters (such as SVD reinitialization frequency) to achieve the best balance of memory efficiency and performance in different tasks?
>
> **Reply:** Thanks for this suggestion. We would like to include a detailed analysis of the influence of each hyperparamter:
>
>
> - **SVD reinitialization frequency**, i.e., number of iterations per round $T_2$:
>     We conducted an additional experiment on IWLST'14 with vanilla transformer to evaluate the impact of the number of iterations, using dimension = 64, $r=8$:
>
>     | Number of iterations per round   | 1     | 2     | 4     | 8     | 16    | 32    |
>     |----------------------------------|-------|-------|-------|-------|-------|-------|
>     | **BLEU**                         | 22.28 | 22.21 | 22.24 | 22.28 | 22.30 | 22.37 |
>
>     The results show that different numbers of iterations yield comparable performance. In all our experiments, we did not tune this hyperparameter and simply fixed it to $d/r$.
>
> - **Iteration interval $T_3$**, i.e., the number of steps per iteration:
>     The influence of the iteration interval has been analyzed in Table 14 in the Appendix, where we show that either very large or very small intervals can be harmful. A large interval causes SST to degrade to LoRA, as the model has limited opportunities to explore new directions of singular vectors. Conversely, a small interval leads to frequent resets of the optimizer states, which can disrupt learning stability.
>
>     In practice, setting the interval to 200 steps strikes a good balance between these extremes. This configuration allows the optimizer to accumulate sufficient states while leaving room for resampling new directions of singular vectors.
>
> - **Rank $r$**, i.e., the low-rank dimension:
>     We view the rank $r$ less as a hyperparameter to tune and more as a constraint determined by available resources. Higher ranks generally improve performance but at the cost of increased memory consumption. In tasks where memory is a critical concern, $r$ should be chosen based on the trade-off between resource limitations and desired performance.
>
>     To evaluate the impact of rank, we conducted an additional experiment on IWSLT'14 using a vanilla Transformer with a model dimension of 128. The results for different methods under varying ranks are shown below:
>
>     | r    | 1     | 2     | 4     | 8     | 16    | 32    | 64    |
>     |------|-------|-------|-------|-------|-------|-------|-------|
>     | LoRA | 12.44 | 14.16 | 16.37 | 20.56 | 23.30 | 25.12 | 26.11 |
>     | ReLoRA | 14.53 | 15.39 | 18.00 | 20.61 | 22.92 | 24.15 | 25.25 |
>     | SST  | 17.49 | 20.69 | 22.80 | 24.19 | 25.12 | 26.08 | 26.15 |
>
>     The evaluation metric is BLEU, where higher scores indicate better performance. The BLEU score for full-rank training is 25.79. The results show that as the rank increases, the performance of all low-rank methods improves. SST consistently outperforms other low-rank methods, particularly at smaller ranks.
>
>     For practical tuning, we recommend starting with $r \approx d/8$ for a good balance between memory efficiency and performance. Increasing $r$ can further improve performance if resources allow.
>
> - **Learning rate**:
>     SST typically performs better with a higher learning rate than full-rank training. However, the learning rate used for full-rank training is also a good starting point for tuning.
>
> - **Warmup steps at the beginning of each iteration**:
>     SST resets optimizer states for $U$, $V^T$, and $\Sigma$ at the start of each iteration. A brief warmup period of around 20 steps, guided by the Adam optimizer’s $\beta$, is recommended to ensure stable training and smooth convergence.
>
> Thanks for your suggestion. We hope this will be helpful for future usage of SST. We have added this in the revised manuscript.

---

> ### Author Response · Authors · 2024-11-28
> **Follow-Up on Revised Manuscript and Looking for forward feedback**
>
> Dear Reviewer TBSo,
>
> We are glad to inform you that we have integrated all your insightful suggestions into the revised manuscript, with the updates highlighted in red for your convenience.
>
> We sincerely thank you for your valuable suggestions, which have greatly helped us improve the quality of our work. We look forward to receiving your feedback at your earliest convenience, as it would allow us sufficient time to conduct any additional experiments or provide further clarifications if needed. Your insights have been extremely helpful, and we deeply appreciate your time and effort in reviewing our work.
>
> Best regards,
>
> The Authors

---

> ### Author Response · Authors · 2024-12-01
> **Update on New Results for LLaMA-7B and Looking for Forward Feedback**
>
> Dear Reviewer TBSo,
>
> We hope this message finds you well. We would like to inform you that we have posted a new **Global Reply 4** to further address the scalability of SST. In this update, we include results from an ongoing experiment with **LLaMA-7B**. The results so far show that SST achieves lower perplexity (PPL) compared to full-rank training up to 35,000 steps for LLaMA-7B. These findings further highlight the effectiveness of SST at larger scales.
>
> We deeply appreciate the valuable feedback you have provided so far, which has been instrumental in refining our work. At this stage, we kindly seek your further feedback or questions regarding our latest results and revisions. Your insights would be invaluable in ensuring the quality and robustness of our submission, and we would be happy to conduct additional experiments or provide clarifications if needed.
>
> Thank you once again for your time and effort in reviewing our work. We look forward to receiving your valuable feedback.
>
> Best regards,
> *Authors*

---

> > ### Author Response · Authors · 2024-12-02
> > **Update on LLaMA-7B Results**
> >
> > Dear Reviewer TBSo,
> >
> > We would like to inform you that we have updated the results of **LLaMA-7B** to **40,000 training steps**. SST continues to surpass full-rank training in perplexity (PPL) at this stage, further demonstrating its effectiveness on larger models.
> >
> > We appreciate your feedback and welcome any additional comments or questions. Thank you for your time and support.
> >
> > Best regards,
> > *The authors of Submission 6242*

---

### Official Review · Reviewer_MMzE · 2024-11-02

**Soundness:** 2
**Presentation:** 2
**Contribution:** 2
**Rating:** 5
**Confidence:** 4

**Summary:**

The paper introduces sparse spectral training (SST) to optimize memory usage for pre-training. SST updates all singular values and selectively updates singular vectors through a multinomial sampling method weighted by the magnitude of the singular values, and employs singular value decomposition to initialize and periodically reinitialize low rank parameters, therefore reducing distortion. The author of the paper conduct experiments on natural language generation, machine translation, node classification, link prediction, and image classification, and SST demonstrates its ability to outperform existing memory reduction training methods and is comparable to full-rank training in various cases.

**Strengths:**

1. This paper is the first to propose parameter-efficient pre-training process in hyperbolic space, in which the design of SST is quite novel and effective.

2. Using SVD to approximate low-rank computation is somehow novel especially with the design of selective updates of singular vector and iterative reinitialization during training.

3. The experimental results show promising performance of SST.

4. This paper is well organized with clear discussion on the disadvantages of the baseline methods, such as ReLoRA.

**Weaknesses:**

1. The overall framework and algorithm design are similar to ReLoRA, which limits the novelty aspect of the paper.

2. Trainable parameter count does not showing significant advantages over baseline methods, such as ReLoRA, and the GPU memory consumption is much larger compared with other methods. Considering the design is using SVD, this weakness is inevitable. Therefore, it limits the wide adoption of this method in the future.

3. Although the author of the paper discusses why the SVD decomposition approach is important, the discussion fails to convince the reader that this approach is necessary. According to the experimental results, ReLoRA achieves quite comparable results with lower memory consumption.

**Questions:**

Please refer to weaknesses.

---

> ### Author Response · Authors · 2024-11-22
> **Reply to the Reviewer MMzE (1/2)**
>
> We greatly appreciate your thoughtful feedback and the opportunity to address your concerns.
>
> **Weakness 1:** The overall framework and algorithm design are similar to ReLoRA, which limits the novelty aspect of the paper.
>
> **Reply:** We apologize if this was unclear in the previous version of the manuscript. **SST fundamentally differs from ReLoRA in both its design philosophy and algorithmic implementation.** As shown in Algorithm 1 (ReLoRA*), the paradigm of ReLoRA, COLA, and PLoRA can be summarized as iteratively performing initialization, update, and merging steps. In contrast, SST eliminates both the initialization and merging steps, replacing them with sampling and swapping (Figure 3). Sampling from existing singular vectors greatly benefits pre-training by avoiding the need to start from random or zero-initialized low-rank matrices.
>
> Below, we outline the key differences and contributions of SST compared to ReLoRA:
>
>
> 1. **Using sampling instead of random or zero initializing:**
>    SST replaces ReLoRA’s initialization step with a sampling mechanism, avoiding the need to start from random or zero-initialized low-rank matrices. This approach leverages existing singular vectors, allowing SST to begin with a more informed and optimized starting point. By doing so, SST enhances exploration and eliminates inefficiencies caused by random initialization. Moreover, SST’s sampling strategy circumvents the saddle point issues observed in ReLoRA, which arise from zero-initialized low-rank components (Figure 2). This ensures smoother and more stable optimization during pre-training.
>
>
> 2. **Dynamic Updates of All Singular Values:**
>    SST updates all singular values at each step, allowing the weight matrix to dynamically adapt. This is particularly effective for pre-training, enabling exploration across the full parameter space. In contrast, ReLoRA could only update within a low-rank subspace in each step.
>
> 3. **Using Singular Value Decomposition to Initialize and Reinitialize Low-Rank Parameters During Training:**
>    SST leverages singular value decomposition (SVD) to initialize low-rank parameters and to periodically reinitialize them during training. SVD guarantees the orthogonality of low-rank matrices, which naturally diminishes over time. By maintaining orthogonality, SST prevents the model from degenerating into a low-rank subspace, preserving its full expressive capacity and ensuring robust exploration throughout the pre-training process. In contrast, ReLoRA performs no such initialization and reinitialization during training.
>
> 4. **Enhanced Gradient Optimization:**
>    SST decouples the learning of direction and magnitude for singular vectors, ensuring that even singular vectors with smaller singular values receive meaningful updates. This balanced optimization fosters better exploration of the spectral domain.
>
>
> These innovations make SST a fundamentally different approach from ReLoRA, specifically designed to address the challenges of pre-training.

---

> ### Author Response · Authors · 2024-11-22
> **Reply to the Reviewer MMzE (2/2)**
>
> **Weakness 2:** Trainable parameter count does not showing significant advantages over baseline methods, such as ReLoRA, and the GPU memory consumption is much larger compared with other methods. Considering the design is using SVD, this weakness is inevitable. Therefore, it limits the wide adoption of this method in the future.
>
>
> **Reply:** Thanks for raising this point. We recognize that our original manuscript lacked a joint consideration of memory reduction and performance. SST's GPU memory consumption is comparable to ReLoRA*, and it reduces the PPL gap by at least 50\% across all pretraining tasks compared to ReLoRA*. A comparison of memory reduction and PPL increasement is in https://anonymous.4open.science/r/SST-iclr2025-review-2DB3/memory_vs_performance.pdf.
>
> Here:
> - **Memory Reduction (%)** = $(\text{Full memory} - \text{Low rank memory}) / \text{Full memory}$,
> - **PPL Increase (\%)** = $(\text{Low rank PPL} - \text{Full PPL}) / \text{Full PPL}$.
>
>
> To better illustrate SST's advantage, we introduced a new metric called **efficiency ratio**, which is defined as:
>
> $ \text{Efficiency Ratio} = \frac{\text{Memory Reduction (\\%)}}{\text{PPL Increase (\\%)}} $
>
>
> **This efficiency ratio represents how much percentage of memory can be reduced at the cost of a 1\% PPL increase.** A higher efficiency ratio indicates a more memory-efficient method.
>
> To clarify, SST achieves a much higher efficiency ratio than ReLoRA*. As shown in https://anonymous.4open.science/r/SST-iclr2025-review-2DB3/memory_vs_performance_ratio.pdf, SST improves the efficiency ratio by:
> - **167.4\%** (OpenWebText, LLaMA-130M)
> - **99.7\%** (C4, LLaMA-130M)
> - **196.1\%** (OpenWebText, OPT-125M)
> - **142.3\%** (OpenWebText, OPT-350M)
> - **65.9\%** (OpenWebText, OPT-1.3B)
> - **4434.3\%** (OpenWebText, LLaMA-1.3B). We put it in the bottom of plot.
>
> These results demonstrate that SST provides a substantially better trade-off between memory reduction and PPL increase compared to ReLoRA*.
>
> We also want to thank the reviewer for highlighting the importance of jointly evaluating memory reduction and PPL increase. This feedback helped us refine the presentation of our findings, and we hope the new metric provides additional clarity on SST's strengths. We have added this to our revised manuscript.
>
>
> **Weakness 3:** Although the author of the paper discusses why the SVD decomposition approach is important, the discussion fails to convince the reader that this approach is necessary. According to the experimental results, ReLoRA achieves quite comparable results with lower memory consumption.
>
> **Reply:** Thank you for your feedback. Please see the reply that we reported above to address the Weakness 2. SST offers a substantially better efficiency ratio compared to ReLoRA*. These results highlight the necessity and advantages of SST's SVD-based approach for pretraining.
>
> In Table 12 of our ablation study, we show that SST, when using random initialization for trainable low-rank matrices (similar to ReLoRA*), achieves a significantly lower BLEU score of 16.03 compared to the original SST's 22.80. This highlights the importance of SST's core idea—using sampling of singular vector instead of random initialization—for effective training from scratch.
>
> |                        | LoRA   | ReLoRA* | SST - random initializing | SST       |
> |------------------------|--------|---------|---------------------------|-----------|
> | BLEU                   | 16.37  | 18.00   | 16.03                     | **22.80** |

---

> ### Author Response · Authors · 2024-11-28
> **Follow-Up on Revised Manuscript and Looking for forward feedback**
>
> Dear Reviewer MMzE,
>
> We are pleased to share that we have incorporated all your valuable suggestions into the revised manuscript, with the changes clearly marked in red for ease of review.
>
> Your feedback has been instrumental in improving the quality of our work, and we sincerely appreciate your thoughtful input. We would be grateful for any further feedback you might have at your earliest convenience, as it will allow us to have sufficient time to address additional questions or conduct further experiments if needed. Thank you once again for your time and effort in reviewing our work.
>
>
> Best regards,
>
> The Authors

---

> ### Author Response · Authors · 2024-12-01
> **Update on New Results for LLaMA-7B and Looking for Forward Feedback**
>
> Dear Reviewer MMzE,
>
> We hope this message finds you well. We would like to inform you that we have posted a new **Global Reply 4** to further address the scalability of SST. In this update, we include results from an ongoing experiment with **LLaMA-7B**. The results so far show that SST achieves lower perplexity (PPL) compared to full-rank training up to 35,000 steps for LLaMA-7B. These findings further highlight the effectiveness of SST at larger scales.
>
> We deeply appreciate the valuable feedback you have provided so far, which has been instrumental in refining our work. At this stage, we kindly seek your further feedback or questions regarding our latest results and revisions. Your insights would be invaluable in ensuring the quality and robustness of our submission, and we would be happy to conduct additional experiments or provide clarifications if needed.
>
> Thank you once again for your time and effort in reviewing our work. We look forward to receiving your valuable feedback.
>
> Best regards,
> *Authors*

---

> > ### Author Response · Authors · 2024-12-02
> > **Update on LLaMA-7B Results**
> >
> > Dear Reviewer MMzE,
> >
> > We would like to inform you that we have updated the results of **LLaMA-7B** to **40,000 training steps**. SST continues to surpass full-rank training in perplexity (PPL) at this stage, further demonstrating its effectiveness on larger models.
> >
> > We appreciate your feedback and welcome any additional comments or questions. Thank you for your time and support.
> >
> > Best regards,
> > *The authors of Submission 6242*

---

### Official Review · Reviewer_cEiN · 2024-11-03

**Soundness:** 2
**Presentation:** 3
**Contribution:** 2
**Rating:** 5
**Confidence:** 3

**Summary:**

This paper propose a novel memory-efficient LLM pre-training method SST, which employs singular value decomposition to initialize and periodically reinitialize lowrank parameters. The experimental results illustrate that the proposed can achieve a better performance than LoRA and its variants.

**Strengths:**

1. The paper focused on an important problem about memoey-efficient LLM pre-training.

2. The proposed memory-efficient LLM pre-training method SST is easy to follow.

3. This paper provides diverse results on different tasks to verify the performance of proposed method. The results illustrate that the proposed method can achieve a better performance than LoRA and its variants.

**Weaknesses:**

1. The motivation and intuition is not very clear. I'm not very clear about the reason why we need to use the proposed method Sparse Spectral Training (SST) for pre-training.  I mean why we need to leverage sparse updates within the spectral domain of neural network
weights. I understand the paper  provided some limitations of LoRA ReLoRA and Galore, but some related work tried to solve these limitations. For example, the limitation of ReLoRA about zero initialization of B can result in zero gradients of A, some related work has solved it, such as Pissa [1].

2. The additional training cost from SST could be increased with scaling the model size. For example, the results of Table 16 about overall training time shows the training time will be increased from 303.4h to 387.2h when using SST.

3. The sensitivity analysis of the proposed method about hyper-parameters, such as the number of iterations and iteration interval.

4. The experiments for pre-training task mainly focus on the small scale, not sure the results when sclaing to a larger dataset and model.

[1] PISSA: PRINCIPAL SINGULAR VALUES AND SINGULAR VECTORS ADAPTATION OF LARGE LANGUAGE MODELS

**Questions:**

1. I would like to ask whether the authors tries to tune the hyper-parameters of baseline methods, such as learning rate, rank value.

---

> ### Author Response · Authors · 2024-11-22
> **Reply to the Reviewer cEiN (1/4)**
>
> Thank you for your insightful comments and suggestions. We are grateful for the opportunity to clarify and improve our work.
>
> **Weakness 1:** The motivation and intuition is not very clear...such as Pissa [1].
>
> **Reply:** We apologize if this was unclear in the previous version of the manuscript. The sparse update of singular vectors is critical for pre-training tasks. PiSSA, which aims for fine-tuning, performs singular value decomposition (SVD) on pre-trained weights and updates the singular vectors with the largest singular values, which is equivalent to replacing SST's sampling strategy with always selecting the top-$r$ singular vectors. To compare these approaches, we conducted an experiment on IWSLT'14 using a vanilla Transformer with a dimension of 64 and $r = 8$:
>
> |      | LoRA | ReLoRA* | SST       | SST always update top-r singular vectors |
> |--|--|--------|---|---|
> | BLEU   |  18.08   | 18.12   | **22.28** | 18.28  |
>
> The evaluation metric is BLEU, where higher scores indicate better performance. The results show that if SST always updates the top-$r$ singular vectors in the weight matrix, its performance becomes similar to LoRA and ReLoRA*. This is expected during pre-training, where the weight matrix is initialized with random values. Always updating the top-$r$ singular vectors effectively reduces SST to the behavior of LoRA, as the same principal components remain dominant in subsequent iterations. This limits exploration, causing the search space to degrade into a low-rank subspace, which is detrimental for pre-training but acceptable for fine-tuning. **These findings demonstrate that sparse updates are crucial for pre-training, as they encourage the model to explore diverse directions, as discussed in Section 4.4.** We thank reviewer for raising this point and we have included this as additional ablation test in revised manuscript.
>
> Additionally, we conducted experiments to compare other different sampling mechanisms:
>
> |                | MULTINOMIAL | UNIFORM | SEQUENTIAL | TOP_R |
> |---|---|---|---|---|
> | BLEU           | **22.28**   | 22.01   | 22.13      | 18.28   |
>
> Here:
> - MULTINOMIAL: The multinomial random sampling method used in SST.
> - UNIFORM: Uniform random sampling.
> - SEQUENTIAL: Iterating through all singular vectors without repetition.
> - TOP_R: Selecting the top-$r$ singular vectors with the largest singular values.
>
> TOP_R performs the worst due to its restricted low-rank subspace, while the other methods show comparable performance, with MULTINOMIAL having a slight advantage.
>
> We would also like to highlight that SST incorporates several additional innovations beyond sparse updates compared to PiSSA. This includes:
>
> - **Updating all singular values.** Unlike PiSSA, which updates only a subset of singular values corresponding to the largest singular vectors, SST updates all singular values at each step. This approach enables SST to dynamically adjust the importance of each singular vector, allowing the model to better explore diverse directions in the spectral domain. This is particularly effective for pre-training, where the goal is to train from scratch and comprehensively capture meaningful representations across the entire parameter space, rather than operating within the constraints of a fixed low-rank subspace as in fine-tuning.
>
> - **Periodic re-SVD.** While PiSSA performs SVD only at initialization, SST periodically recomputes the SVD during pre-training. Over time, the orthogonality among the singular vectors in $U$ and $V^T$ tends to diminish, which can lead to a degenerated low-rank subspace and reduced expressive power. Periodic re-SVD ensures that the orthogonality of these singular vectors is preserved, maintaining the model’s capacity to explore diverse directions in the spectral domain. This is important for pre-training, where continuous exploration of new directions is necessary for effective convergence.
>
> - **Enhance gradient of U and V.** SST introduces an enhanced gradient calculation for the singular vectors $U$ and $V$ in the sparse spectral layer. Unlike the default gradient approach, which scales the updates for $U$ and $V$ by the corresponding singular values, SST decouples the learning of the singular vectors’ directions from their magnitudes. This adjustment ensures that singular vectors with smaller singular values still receive meaningful gradient updates, rather than being overshadowed by those with larger singular values.
>
> Finally, we want to thank the reviewer for mentioning PiSSA. We became aware of PiSSA after its release at NeurIPS and we have included it in the related work section of the revised manuscript. It is noteworthy that PiSSA is designed primarily for fine-tuning, while SST is specifically tailored for pre-training tasks. We are pleased to see that both methods independently leverage the spectral domain, underscoring the value of SVD-based approaches in advancing low-rank memory-efficient training methods.

---

> ### Author Response · Authors · 2024-11-22
> **Reply to the Reviewer cEiN (2/4)**
>
> **Weakness 2:** The additional training cost from SST could be increased with scaling the model size. For example, the results of Table 16 about overall training time shows the training time will be increased from 303.4h to 387.2h when using SST.
>
> **Reply:** Thank you for raising this concern. As shown in Table 16, SST’s overall training time is slightly longer compared to LoRA and ReLoRA for the same number of training steps: by 0.9\% for OPT-125M, 10.9\% for OPT-350M, and 19.2\% for OPT-1.3B. However, according to Figure 4, SST achieves superior performance even with fewer training steps. To illustrate this, we compared the perplexity (PPL) on the validation set for SST trained with 20\% fewer steps than other methods:
>
> | Model      | Full   | LoRA   | ReLoRA* | SST (with 20\% less step) |
> |------------|--------|--------|---------|---------------------------|
> | OPT-125M   | 23.50  | 34.23  | 35.80   | **28.03**                     |
> | OPT-350M   | 21.78  | 34.26  | 39.21   | **29.42**                     |
> | OPT-1.3B   | 15.10  | 1716   | 29.52   | **22.98**                     |
> | LLaMA-130M | 20.04  | 29.71  | 31.33   | **24.74**                     |
> | LLaMA-1.3B | 14.54  | 16.50  | 17.32   | **15.65**                     |
>
> These results demonstrate that SST maintains significantly lower perplexity even with fewer training steps, highlighting its efficiency. While SST incurs a marginally higher computational cost per step, its ability to achieve superior performance with reduced training steps compensates for this increase, making it a more effective approach for achieving high-quality pretraining. We will add these findings to the revised manuscript.
>
>
> **Weakness 3:** The sensitivity analysis of the proposed method about hyper-parameters, such as the number of iterations and iteration interval.
>
> **Reply:** Thank you for this suggestion. The influence of the iteration interval is already analyzed in Table 14 in the Appendix, where we demonstrate that either a very large or very small iteration interval can be harmful. A large interval causes SST to degrade to LoRA, while a small interval results in overly frequent resets of the optimizer states. However, in our experiments, we did not tune the iteration interval. Simply setting the interval to 200 was sufficient in most scenarios, as it balances the accumulation of optimizer states and leaving space for enough iterations of resampling for exploring new directions of singular vectors. We apologize for any confusion caused by the terminology in Table 14, where we referred to "the number of steps per iteration", which is equivalent to "iteration interval". In the revised manuscript, we have standardized the terminology by replacing "the number of steps per iteration" with "iteration interval".
>
> We also conducted an additional experiment on IWLST'14 with vanilla transformer to evaluate the impact of the number of iterations, using dimension = 64, $r=8$:
>
> |  Number of iterations per round   | 1     | 2     | 4     | 8     | 16    | 32    |
> |---------------------|-------|-------|-------|-------|-------|-------|
> | BLEU                | 22.28 | 22.21 | 22.24 | 22.28 | 22.30 | 22.37 |
>
> The results show that different numbers of iterations yield comparable performance. In our experiments, we did not tune this hyperparameter either, and simply fixed it to $d/r$. We have added this in the revised manuscript.

---

> ### Author Response · Authors · 2024-11-22
> **Reply to the Reviewer cEiN (3/4)**
>
> **Weakness 4:** The experiments for pre-training task mainly focus on the small scale, not sure the results when sclaing to a larger dataset and model.
>
> **Reply:** Thank you for raising this point. We address this concern in the Global Reply, where we detail additional experiments conducted on larger datasets and models.
>
> To make our findings clearer, we summarize the experiments as follows:
>
> 1. **Scaling to Larger Datasets (C4):**
>    - The C4 dataset, which is 25 times larger than OpenWebText, was used to evaluate SST’s performance on a significantly larger dataset.
>    - We want to highlight that even OpenWebText is a very large dataset, containing approximately 9 billion tokens. For a 1B-parameter model, which is pre-trained with 13.1 billion tokens, the pre-training process could only cover one epoch of OpenWebText.
>    - The results show that SST reduces the perplexity gap by **16.4%** on the C4 dataset compared to other low-rank methods.
>
> 2. **Scaling to Larger Models (LLaMA-1.3B):**
>    - We extended our experiments to LLaMA-1.3B.
>    - The results demonstrate that SST outperforms other low-rank methods like LoRA and ReLoRA* on LLaMA-1.3B, achieving a PPL that closely matches full-rank training. SST reduces the perplexity gap to full-rank by **97.4%**.
>
> The final PPL results for these experiments are summarized below (bold indicates the lowest PPL among all low-rank methods):
>
> | Dataset       | Model        | r/d       | Full lr=1e-3 | LoRA lr=1e-3 | ReLoRA* lr=1e-3 | SST lr=1e-3 | LoRA lr=3e-3 | ReLoRA* lr=3e-3 | SST lr=3e-3 |
> |---------------|--------------|-----------|--------------|--------------|-----------------|-------------|--------------|-----------------|-------------|
> | C4            | LLaMA-130M  | 64/768    | 24.91        | 35.91        | 37.34           | 32.13       | 30.75        | 133.06          | **29.79**   |
> | OpenWebText   | LLaMA-130M  | 64/768    | 20.04        | 29.71        | 31.33           | 25.89       | 795.24       | 230.43          | **23.35**   |
>
> | Dataset       | Model      | r/d        | Full lr=4e-4 | LoRA lr=1e-3 | ReLoRA* lr=1e-3 | SST lr=1e-3 |
> |---------------|------------|------------|--------------|--------------|-----------------|-------------|
> | OpenWebText   | LLaMA-1.3B | 128/2048   | 14.54        | 16.50        | 17.32           | **14.59**   |
>
> Each method was trained with 2.6 billion tokens for LLaMA-130M and 13.1 billion tokens for LLaMA-1.3B. These findings demonstrate SST’s scalability and effectiveness when applied to larger datasets and models, further supporting its use as a robust memory-efficient training method. Thanks reviewer for raising this point. We will include these new experiments in the updated version of the article.

---

> ### Author Response · Authors · 2024-11-22
> **Reply to the Reviewer cEiN (4/4)**
>
> **Question 1:** I would like to ask whether the authors tries to tune the hyper-parameters of baseline methods, such as learning rate, rank value.
>
> **Reply:** Thanks for raising this point. Below, we provide a detailed explanation of how we handled hyper-parameters tuning of baseline methods in our experiments:
>
> - **Learning rate**: LoRA is primarily designed for fine-tuning, and ReLoRA’s original version includes an initial period of full-rank training, making it difficult to directly adopt their original learning rates. To ensure fair comparisons, we used the same learning rate for LoRA, ReLoRA*, and SST in all experiments, equal to or higher than the learning rate for full-rank training. The detailed learning rates for all methods in all experiments are provided in Appendix D. In our Global Reply, we discussed that SST achieves lower PPL than LoRA and ReLoRA* on LLaMA-130M with both $lr = 1\mathrm{e}{-3}$ and $lr = 3\mathrm{e}{-3}$ on both C4 and OpenWebText.
>
>     The final PPL for LLaMA-130M on C4 and OpenWebText datasets is summarized below (bold indicates the lowest PPL among all low-rank methods):
>
>     | Dataset       | Model        | r/d       | Full lr=1e-3 | LoRA lr=1e-3 | ReLoRA* lr=1e-3 | SST lr=1e-3 | LoRA lr=3e-3 | ReLoRA* lr=3e-3 | SST lr=3e-3 |
>     |---------------|--------------|-----------|--------------|--------------|-----------------|-------------|--------------|-----------------|-------------|
>     | C4            | LLaMA-130M  | 64/768    | 24.91        | 35.91        | 37.34           | 32.13       | 30.75        | 133.06          | **29.79**   |
>     | OpenWebText   | LLaMA-130M  | 64/768    | 20.04        | 29.71        | 31.33           | 25.89       | 795.24       | 230.43          | **23.35**   |
>
>     Each method was trained with 2.6 billion tokens. The learning rate of 1e-3 for full-rank training matches the setting in Table 1 of the ReLoRA article. SST achieves lower PPL than LoRA and ReLoRA* at the same learning rate on both datasets, and with $lr=3\mathrm{e}{-3}$, SST surpasses all other low-rank methods, reducing the perplexity gap by **16.4%** on C4 and **65.8%** on OpenWebText.
>
>
>
>
>
>
> - **Rank**: For all low-rank methods, including LoRA, ReLoRA*, and SST, rank is more of a constraint determined by available resources than a hyperparameter to be tuned. Higher ranks generally improve performance but at the cost of increased memory consumption. To ensure fairness, we used the same rank values for LoRA, ReLoRA*, and SST in all experiments, as these methods have similar amount of trainable parameters under same rank.
>
>     Additionally, we conducted an experiment on IWSLT'14 with a vanilla Transformer (dimension = 128) to analyze the impact of rank on different methods:
>
>     | Rank ($r$) | 1     | 2     | 4     | 8     | 16    | 32    | 64    |
>     |------------|-------|-------|-------|-------|-------|-------|-------|
>     | **LoRA**   | 12.44 | 14.16 | 16.37 | 20.56 | 23.30 | 25.12 | 26.11 |
>     | **ReLoRA** | 14.53 | 15.39 | 18.00 | 20.61 | 22.92 | 24.15 | 25.25 |
>     | **SST**    | 17.49 | 20.69 | 22.80 | 24.19 | 25.12 | 26.08 | 26.15 |
>
>     The evaluation metric is BLEU, where higher scores indicate better performance. The BLEU score for full-rank training is 25.79. The results show that as the rank increases, the performance of all methods improves. SST consistently outperforms other low-rank methods, particularly at smaller ranks.
>
>
> - **Scale factor $\alpha$ in GaLore**: We evaluated both $\alpha = 0.25$ (as used in GaLore) and $\alpha = 1$ (no scaling) in our Transformer and OPT experiments. As shown in Appendix F, $\alpha = 0.25$ consistently underperforms compared to $\alpha = 1$ in our experiments.
>
> - **Iteration interval of ReLoRA*** (written as reset interval in ReLoRA article): We used the same reset interval for ReLoRA* as the iteration interval for SST, as both methods reset optimizer states at this step.
>
>     The influence of the iteration interval is discussed in Table 14 in the Appendix. We show that very large intervals reduce opportunities to explore new directions of singular vectors, while very small intervals result in overly frequent resets of the optimizer states, disrupting learning stability. This reasoning applies to ReLoRA* as well.

---

> ### Author Response · Authors · 2024-11-28
> **Follow-Up on Revised Manuscript and Looking for forward feedback**
>
> Dear Reviewer cEiN,
>
> We wanted to kindly inform you that we have carefully incorporated all the suggestions and feedback you shared into the revised manuscript. For your convenience, the revisions have been marked in red.
>
> We are deeply grateful for your thoughtful suggestions, which have significantly enhanced the quality of our work. We would greatly appreciate your feedback at your earliest convenience, as it would give us sufficient time to address any further questions or conduct additional experiments if necessary. Your insights have been invaluable, and we truly appreciate your time and effort in reviewing our manuscript.
>
>
> Best regards,
>
> The Authors

---

> > ### Comment · Reviewer_cEiN · 2024-12-03
> > **Thank you very much for your response**
> >
> > Thank you for your response and efforts to address my concerns. My primary concerns center on the experiment and hyperparameter selection.
> >
> > (1) The persistent performance gap between SST and full-rank training raises questions about the method's applicability in practice.
> >
> > (2) Regarding the trade-off between low-rank and high-rank training, while higher ranks can indeed narrow the performance gap, I notice that the improvements become marginal at larger rank values (e.g., 26.11 vs 26.15 for LoRA vs SST at rank 64). This observation prompts questions about performance at even larger rank values, such as 128 or 256. It would be valuable to understand whether SST maintains its advantage over LoRA in these higher-rank scenarios.

---

> ### Author Response · Authors · 2024-12-01
> **Update on New Results for LLaMA-7B and Looking for Forward Feedback**
>
> Dear Reviewer cEiN,
>
> We hope this message finds you well. We would like to inform you that we have posted a new **Global Reply 4** to further address the scalability of SST. In this update, we include results from an ongoing experiment with **LLaMA-7B**. The results so far show that SST achieves lower perplexity (PPL) compared to full-rank training up to 35,000 steps for LLaMA-7B. These findings further highlight the effectiveness of SST at larger scales.
>
> We deeply appreciate the valuable feedback you have provided so far, which has been instrumental in refining our work. At this stage, we kindly seek your further feedback or questions regarding our latest results and revisions. Your insights would be invaluable in ensuring the quality and robustness of our submission, and we would be happy to conduct additional experiments or provide clarifications if needed.
>
> Thank you once again for your time and effort in reviewing our work. We look forward to receiving your valuable feedback.
>
> Best regards,
> *Authors*

---

> > ### Author Response · Authors · 2024-12-02
> > **Update on LLaMA-7B Results**
> >
> > Dear Reviewer cEiN,
> >
> > We would like to inform you that we have updated the results of **LLaMA-7B** to **40,000 training steps**. SST continues to surpass full-rank training in perplexity (PPL) at this stage, further demonstrating its effectiveness on larger models.
> >
> > We appreciate your feedback and welcome any additional comments or questions. Thank you for your time and support.
> >
> > Best regards,
> > *The authors of Submission 6242*

---

> ### Author Response · Authors · 2024-12-04
> **Additional Reply to Reviewer cEiN**
>
> Thanks you for your response. We are grateful for the opportunity to clarify your concerns.
>
> **Concern 1:** The persistent performance gap between SST and full-rank training raises questions about the method's applicability in practice.
>
> **Reply:**
>
> 1. Experiments on LLaMA-1.3B and LLaMA-7B demonstrate that SST is comparable to full-rank training, even with **low rank**. For example:
>    - On LLaMA-1.3B, using rank $r = 128$, equal to **1/16** of the model dimension $d = 2048$, SST achieves a PPL of **14.59** vs **14.54** for full-rank training.
>    - On LLaMA-7B, with rank $r = 256$, equal to **1/16** of $d = 4096$, until 40K steps, SST achieves a PPL of **19.66** vs **19.76** for full-rank training.
>
> 2. In experiments on Euclidean and Hyperbolic Transformers, SST is also better than, full-rank training in some configurations, even with rank values less than or equal to **1/8** of the original model dimension.
>
>
>
>
> **Concern 2:** Regarding the trade-off between low-rank and high-rank training, while higher ranks can indeed narrow the performance gap, I notice that the improvements become marginal at larger rank values (e.g., 26.11 vs 26.15 for LoRA vs SST at rank 64). This observation prompts questions about performance at even larger rank values, such as 128 or 256. It would be valuable to understand whether SST maintains its advantage over LoRA in these higher-rank scenarios.
>
>
> **Reply:**
> 1. We believe the reviewer is referring to the following table:
>
>     | Rank ($r$)                               | 1     | 2     | 4     | 8     | 16    | 32     | 64    |
>     |------------------------------------------|-------|-------|-------|-------|-------|--------|-------|
>     | **LoRA**                                 | 12.44 | 14.16 | 16.37 | 20.56 | 23.30 | 25.12  | 26.11 |
>     | **ReLoRA**                               | 14.53 | 15.39 | 18.00 | 20.61 | 22.92 | 24.15  | 25.25 |
>     | **SST**                                  | 17.49 | 20.69 | 22.80 | 24.19 | 25.12 | 26.08  | 26.15 |
>     | **SST relative performance improvement** | 26.3% | 50.1% | 61.7% | 69.1% | 73.1% | 143.3% | -     |
>
>     The **SST Relative Performance Improvement** is defined as:
>     $$\frac{\text{SST BLEU} - \max(\text{LoRA BLEU}, \text{ReLoRA BLEU})}{\text{Full BLEU} - \max(\text{LoRA BLEU}, \text{ReLoRA BLEU})}$$
>
>     The BLEU score for full-rank training is 25.79. The entry "-" for $r = 64$ in "SST Relative Performance Improvement" indicates that LoRA surpasses full-rank training, resulting the calculation of relative improvement undefined. The table shows that while the absolute performance gap between methods narrows as rank increases, the **relative performance improvement** of SST grows significantly, from 26.3% to 143.3%.
>
> 2. As noted by the reviewer, "the improvements become marginal at larger rank values (e.g., 26.11 vs 26.15 for LoRA vs SST at rank 64)." In this experiment, the original model has a dimension of 128, meaning $r = 64$ achieves no reduction in trainable parameters (density ratio $\approx 2r/d$), which is not applicable since low-rank methods are designed to reduce trainable parameters. Larger ranks, such as $r = 128$ or $256$, would even lead to an increase in the number of parameters. Naturally, when trainable parameters approach those of full-rank training, all methods are expected to perform similarly.
>
> 3. With $r = 16$ (i.e., **1/8** of the model dimension), SST achieves comparable performance to full-rank training (25.12 vs 25.79 BLEU). This demonstrates why we typically use $r \leq 1/8 d$ in our experiments, to achieve significant parameter reduction without incurring a large degradation in performance.
>
> 4. Our focus on low-rank settings is to make pre-training LLMs more accessible to most researchers in the research community. We believe that achieving comparable performance to full-rank training in low-rank settings is a more challenging and impactful goal than achieving performance parity in high-rank scenarios.

---

### Official Review · Reviewer_3vN4 · 2024-11-08

**Soundness:** 2
**Presentation:** 2
**Contribution:** 3
**Rating:** 5
**Confidence:** 4

**Summary:**

This paper proposes Sparse Spectral Training (SST), a memory-efficient training method designed for large neural networks. SST aims to reduce memory usage while maintaining performance close to full-rank models. The method updates all singular values of network weights and selectively updates singular vectors based on their significance, using a sampling strategy that prioritizes impactful components. SST is evaluated on tasks across Euclidean and hyperbolic neural networks, showing that it often outperforms existing memory reduction techniques and occasionally matches or exceeds full-rank training performance, especially in low-dimensional or resource-constrained settings.

**Strengths:**

Sparse Spectral Training (SST) introduces an interesting approach to memory-efficient training by updating all singular values while selectively focusing on significant singular vectors. This approach shows promise for reducing memory requirements and can be a creative improvement on existing low-rank adaptation methods.

**Weaknesses:**

While SST is promising, the current experiments lack rigor, especially as the performance comparisons rely on relatively high perplexity values and low zero-shot accuracies, limiting the conclusions that can be drawn about SST's effectiveness. For a more meaningful evaluation, it would be valuable to benchmark against models that achieve stronger results (e.g., perplexity below 10). Additionally, the selection of baselines could be strengthened to more comprehensively illustrate SST’s relative performance. These changes would help clarify SST’s potential as a competitive memory-efficient training method.

**Questions:**

- How would SST perform if compared against models capable of achieving lower perplexity scores (e.g., below 10)? What is the extent to which SST’s performance might vary when applied to higher-performing models with more rigorous baseline comparisons?
- Are there alternative sampling mechanisms that the authors considered, and could these impact the efficiency or stability of SST?

---

> ### Author Response · Authors · 2024-11-22
> **Reply to the Reviewer 3vN4 (1/2)**
>
> Thank you for your valuable and constructive feedback. We appreciate the opportunity to address your concerns.
>
> **Weakness 1:** While SST is promising, the current experiments lack rigor, especially as the performance comparisons rely on relatively high perplexity values and low zero-shot accuracies, limiting the conclusions that can be drawn about SST's effectiveness. For a more meaningful evaluation, it would be valuable to benchmark against models that achieve stronger results (e.g., perplexity below 10). Additionally, the selection of baselines could be strengthened to more comprehensively illustrate SST’s relative performance. These changes would help clarify SST’s potential as a competitive memory-efficient training method.
>
> **Reply:** Thank you for this valuable feedback. We understand the reviewer’s concern regarding whether the effectiveness of SST can be generalized to larger models. We agree that a more extensive evaluation against models achieving perplexity below 10 would further strengthen the conclusions. However, pre-training large language models (LLMs) to such low perplexity levels is extremely resource-intensive. As reported in [1], pre-training GPT-3 Medium to achieve a PPL of 10.6 would require approximately 800 days of A100 GPU time.
>
> The primary goal of SST, as a memory-efficient pre-training method, is to lower the resource requirements for pre-training LLMs, making such endeavors more accessible to researchers. Our focus is on demonstrating SST's ability to reduce memory costs and improve efficiency under practical conditions. Considering that models around 1B parameters are the largest most researchers in the community can realistically pre-train due to resource constraints [1,2,3], we selected 1B as the upper limit for our experiments to ensure the results remain relevant and accessible.
>
> **To further address the reviewer’s concerns, we conducted additional experiments pre-training LLaMA models (130M and 1.3B),** which represent a more advanced architecture compared to OPT. With the same number of parameters, LLaMA achieves lower perplexity compared to OPT. The results of these experiments, detailed in the Global Reply, demonstrate that SST consistently outperforms other low-rank methods such as LoRA and ReLoRA* in PPL. SST reduces the perplexity gap by **16.4%** (C4) and **65.8%** (OpenWebText).
>
> The final PPL for LLaMA-1.3B is summarized below (bold indicates the lowest PPL among all low-rank methods):
>
> | Dataset       | Model      | r/d        | Full lr=4e-4 | LoRA lr=1e-3 | ReLoRA* lr=1e-3 | SST lr=1e-3 |
> |---------------|------------|------------|--------------|--------------|-----------------|-------------|
> | OpenWebText   | LLaMA-1.3B | 128/2048   | 14.54        | 16.50        | 17.32           | **14.59**   |
>
> Each method was trained with 13.1 billion tokens. The learning rate for full-rank training is 4e-4, consistent with the configuration in Table 1 of the ReLoRA article. As recommended in Appendix A of the ReLoRA article, the learning rate for ReLoRA should be twice that of full-rank training. Thus, we used a learning rate of 1e-3 for all low-rank methods. SST outperforms all other low-rank methods, closely matching the PPL of full-rank training and reducing the perplexity gap to full-rank by **97.4%**.
>
> However, when training GaLore with the same hyperparameter settings as specified in its article, GaLore encountered instability and crashed with rank 128. The training loss curve has been uploaded for reference at  https://anonymous.4open.science/r/SST-iclr2025-review-2DB3/GaLore.png.
>
> Thanks reviewer for raising this. We will include these new experiments in the updated version of the article.
>
> **The reviewer also mentioned that the selection of baselines could be strengthened.** To the best of our knowledge, the baselines we used—ReLoRA and GaLore—are already state-of-the-art methods in low-rank pre-training of LLMs. If there are other methods the reviewer believes should be included, we are more than willing to consider them and provide additional comparisons where feasible. We greatly appreciate any specific suggestions in this regard.
>
>
>
>
>
>
>
> **Question 1:** How would SST perform if compared against models capable of achieving lower perplexity scores (e.g., below 10)? What is the extent to which SST’s performance might vary when applied to higher-performing models with more rigorous baseline comparisons?
>
> **Reply:** Thank you for your feedback. Please see the reply that we reported above to address the Weakness 1.

---

> ### Author Response · Authors · 2024-11-22
> **Reply to the Reviewer 3vN4 (2/2)**
>
> **Question 2:** Are there alternative sampling mechanisms that the authors considered, and could these impact the efficiency or stability of SST?
>
> **Reply:** Thank you for your insightful suggestion. We conducted additional experiments to analyze the influence of different sampling mechanisms:
>
> |                | MULTINOMIAL | UNIFORM | SEQUENTIAL | TOP_R |
> |----------------|-------------|---------|------------|---------|
> | BLEU           | **22.28**   | 22.01   | 22.13      | 18.28   |
>
> All sampling strategies were trained with SST on a vanilla Transformer with a dimension of 64 and $r = 8$, using the IWSLT'14 dataset. The evaluation metric is BLEU, where higher scores indicate better performance.
>
> - MULTINOMIAL: The multinomial random sampling method used in SST.
> - UNIFORM: Uniform random sampling.
> - SEQUENTIAL: Iterating through all singular vectors without repetition.
> - TOP_R: Selecting the top-$r$ singular vectors with the largest singular values.
>
> We also considered a Binomial sampling mechanism; however, it could not guarantee that the number of selected singular vectors would remain consistent with the specified rank, making it unsuitable for direct comparison.
>
> The results indicate that TOP_R performs the worst, as its search space degenerates into a low-rank subspace. In contrast, as long as all singular vectors are visited, the other methods exhibit comparable performance. Among them, MULTINOMIAL shows a slight advantage.
>
> We appreciate the reviewer for raising this point. These findings have been included in the revised manuscript.
>
>
>
> [1] Dettmers, T., Lewis, M., Shleifer, S., & Zettlemoyer, L. (2022). 8-bit Optimizers via Block-wise Quantization. International Conference on Learning Representations.
> [2] Lialin, Vladislav, et al. "Relora: High-rank training through low-rank updates." The Twelfth International Conference on Learning Representations. 2023.
> [3] Hu, Yuezhou, et al. "Accelerating Transformer Pre-training with 2: 4 Sparsity." Forty-first International Conference on Machine Learning.

---

> ### Author Response · Authors · 2024-11-28
> **Follow-Up on Revised Manuscript and Looking for forward feedback**
>
> Dear Reviewer 3vN4,
>
> We would like to kindly update you that we have incorporated all the suggestions and feedback you provided into the revised manuscript. The revisions are clearly marked in red for your convenience.
>
> We sincerely thank you for your valuable suggestions, which have greatly helped us improve the quality of our work. We look forward to receiving your feedback at your earliest convenience, as it would allow us sufficient time to conduct any additional experiments or provide further clarifications if needed. Your insights have been extremely helpful, and we deeply appreciate your time and effort in reviewing our work.
>
>
> Best regards,
>
> The Authors

---

> ### Author Response · Authors · 2024-12-01
> **Update on New Results for LLaMA-7B and Looking for Forward Feedback**
>
> Dear Reviewer 3vN4,
>
> We hope this message finds you well. We would like to inform you that we have posted a new **Global Reply 4** to further address the scalability of SST. In this update, we include results from an ongoing experiment with **LLaMA-7B**. The results so far show that SST achieves lower perplexity (PPL) compared to full-rank training up to 35,000 steps for LLaMA-7B. These findings further highlight the effectiveness of SST at larger scales.
>
> We deeply appreciate the valuable feedback you have provided so far, which has been instrumental in refining our work. At this stage, we kindly seek your further feedback or questions regarding our latest results and revisions. Your insights would be invaluable in ensuring the quality and robustness of our submission, and we would be happy to conduct additional experiments or provide clarifications if needed.
>
> Thank you once again for your time and effort in reviewing our work. We look forward to receiving your valuable feedback.
>
> Best regards,
> *Authors*

---

> > ### Author Response · Authors · 2024-12-02
> > **Update on LLaMA-7B Results**
> >
> > Dear Reviewer 3vN4,
> >
> > We would like to inform you that we have updated the results of **LLaMA-7B** to **40,000 training steps**. SST continues to surpass full-rank training in perplexity (PPL) at this stage, further demonstrating its effectiveness on larger models.
> >
> > We appreciate your feedback and welcome any additional comments or questions. Thank you for your time and support.
> >
> > Best regards,
> > *The authors of Submission 6242*

---

### Author Response · Authors · 2024-11-22
**Global Reply**

We sincerely thank the reviewers for their valuable and constructive feedback on our submission. Your insights have been very important in enhancing the quality of our work, and we greatly appreciate the time and effort you dedicated to reviewing our paper.

- **In response to Reviewer 3vN4's and Reviewer cEiN's comments on scaling to larger models and achieving lower PPL**:
    We extended our experiments to include LLaMA-1.3B.

    The final PPL for LLaMA-1.3B are summarized below (bold indicates the lowest PPL among all low-rank methods):

    | Dataset       | Model      | r/d        | Full lr=4e-4 | LoRA lr=1e-3 | ReLoRA* lr=1e-3 | SST lr=1e-3 |
    |---------------|------------|------------|--------------|--------------|-----------------|-------------|
    | OpenWebText   | LLaMA-1.3B | 128/2048   | 14.54        | 16.50        | 17.32           | **14.59**   |

    Each method was trained with 13.1 billion tokens. The learning rate for full-rank training is 4e-4, consistent with the configuration in Table 1 of the ReLoRA article. As recommended in Appendix A of the ReLoRA article, the learning rate for ReLoRA should be twice that of full-rank training. Thus, we used a learning rate of 1e-3 for all low-rank methods. SST outperforms all other low-rank methods, closely matching the PPL of full-rank training and reducing the perplexity gap to full-rank by **97.4%**.

- **In response to Reviewer cEiN's comment on using larger datasets and addressing hyperparameter search**:
    We pre-trained LLaMA-130M on the C4 dataset, which is 25 times larger than OpenWebText. To address the comment on hyperparameter tuning, we evaluated each method with two different learning rates.

    The final PPL for LLaMA-130M on C4 and OpenWebText are summarized below (bold indicates the lowest PPL among all low-rank methods):

    | Dataset       | Model        | r/d       | Full lr=1e-3 | LoRA lr=1e-3 | ReLoRA* lr=1e-3 | SST lr=1e-3 | LoRA lr=3e-3 | ReLoRA* lr=3e-3 | SST lr=3e-3 |
    |---------------|--------------|-----------|--------------|--------------|-----------------|-------------|--------------|-----------------|-------------|
    | C4            | LLaMA-130M  | 64/768    | 24.91        | 35.91        | 37.34           | 32.13       | 30.75        | 133.06          | **29.79**   |
    | OpenWebText   | LLaMA-130M  | 64/768    | 20.04        | 29.71        | 31.33           | 25.89       | 795.24       | 230.43          | **23.35**   |

    Each method was trained with 2.6 billion tokens. The learning rate of 1e-3 for full-rank training matches the setting used in Table 1 of the ReLoRA article. We ran LoRA, ReLoRA*, and SST with both $lr=1\mathrm{e}{-3}$ and $lr=3\mathrm{e}{-3}$. The results show that SST achieves lower PPL than LoRA and ReLoRA* for the same learning rate, and with $lr=3\mathrm{e}{-3}$, SST surpasses all other low-rank methods, reducing the perplexity gap by **16.4%** on C4 and **65.8%** on OpenWebText.

---

### Author Response · Authors · 2024-11-22
**Global Reply 2**

Below is a summary of the items we have addressed in our responses to the reviewers, but are currently working on incorporating into the updated manuscript. This ensures that no points have been overlooked by us when revising the manuscript:

- **New Experiments**:
  - Incorporate experiments with LLaMA-130M and 1.3B on OpenWebText and C4 datasets.
  - Include comparisons of different sampling mechanisms, such as always selecting top-$r$ singular vectors.
  - Add the sensitivity analysis for the number of iterations.
  - Add the sensitivity analysis of SST under different rank.
  - Add the experiment to evaluate SST's performance with additional training steps compared to full-rank training.

- **Manuscript Revisions**:
  - Include the plot comparing efficiency ratios.
  - Add the comparison of SST with fewer training steps against other low-rank methods trained with full steps.
  - Add PiSSA to the related work section.
  - Standardize terminology by replacing "the number of steps per iteration" with "iteration interval".

---

### Author Response · Authors · 2024-11-28
**Global Reply 3**

We are pleased to share that all the revisions we committed to in our responses to the reviewers have been successfully incorporated into the updated manuscript. The revised parts are clearly marked in red for ease of review.

We sincerely thank the reviewers for their invaluable feedback and constructive suggestions, which have greatly enhanced the quality of our work.

---

### Author Response · Authors · 2024-11-29
**Request for Assistance Regarding Reviewer Feedback for Submission 6242**

Dear Area Chairs, Senior Area Chairs, and Program Chairs,

We hope this email finds you well. We are the authors of Submission 6242. We posted our replies to all four reviewers 7 days ago, on November 22nd, but have not received any response. We provided another update to our revised manuscript and requested further feedback yesterday, on November 28th, yet there has still been no reply.

We noticed that most other submissions have received responses from their reviewers, and we are concerned that our situation might be due to a technical issue, such as:
1. Reviewers not receiving email notifications about our replies.
2. Reviewer responses being posted but not visible to us.

If this is not due to technical reasons, we kindly request your assistance in notifying the reviewers of our replies. We feel it might be inappropriate for us to send repeated notifications directly to the reviewers, and your support in this matter would be greatly appreciated.

We deeply value the feedback we have received so far, which has significantly helped us improve the quality of our work. Hearing from the reviewers at this stage would allow us to incorporate any additional experiments or provide further clarifications, ensuring the manuscript fully reflects their valuable insights.

Thank you very much for your time and consideration. We sincerely appreciate your efforts in managing the review process.

Best regards,
*The authors of Submission 6242*

---

### Author Response · Authors · 2024-12-01
**Global Reply 4**

Thank you once again to all reviewers for your thoughtful and constructive feedback, which has been instrumental in guiding our revisions. We are pleased to share that we are currently conducting an additional experiment with **LLaMA-7B** to further evaluate the scalability of SST on larger models.

As of now, LLaMA-7B has been trained for 40,000 steps for both SST and full-rank training. Unfortunately, the server for training ReLoRA* crashed, but the training has been restarted.

The perplexity (PPL) for LLaMA-7B on OpenWebText at various training steps is summarized below:

| r/d        | Step    | 5000   | 10000 | 15000 | 20000 | 25000 | 30000 | 35000 | 40000 |
|------------|---------|--------|-------|-------|-------|-------|-------|-------|-------|
| 4096/4096  | Full      | 272.12 | 62.55 | 36.04 | 27.13 | 23.57 | 21.65 | 20.53 | 19.76 |
| 256/4096   | ReLoRA* | 84.41  | 59.04 | 34.50 | 30.23 | 27.69 |  25.31  | 24.11  | 23.54  |
| 256/4096   | SST     | 65.41  | 29.84 | 24.59 | 22.34 | 21.02 | 20.40 | 19.76 | 19.66 |

**(Table updated on December 4th to include results up to 40,000 steps)**

By 40,000 steps, each method has been trained on 5.2 billion tokens. The learning rate for full-rank training is set to 2e-4, which is half the learning rate used for full-rank training in LLaMA-1.3B experiments. Similarly, the learning rate for SST and ReLoRA* is 5e-4, half of the SST and ReLoRA* learning rate used in LLaMA-1.3B. Since the ReLoRA and GaLore articles and codebases do not specify the learning rate for full-rank training of LLaMA-7B, we have followed the trend of reducing learning rates with increasing model size as observed in their articles.

As of 40,000 steps, SST continues to outperform full-rank training in terms of PPL, demonstrating its effectiveness even at larger scales. We will provide further updates as the training progresses and additional results become available.

Thank you for your continued support and valuable suggestions, which have been crucial in shaping this work. We look forward to any further feedback you may have.

---

### Author Response · Authors · 2024-12-04
**Summary for Area Chair and Reviewers of Submission 6242 (1/3)**

Dear Area Chairs and Reviewers,

We are truly grateful for all the effort Area Chair have made to encourage reviewers to respond. We fully understand the difficulty of organizing the review process this year, especially with submissions increasing by 61%. Your dedication has been essential in maintaining the fair and supportive culture of the AI research community.

We would also like to thank all the reviewers for their insightful suggestions, as well as the recommendation to test our approach on larger models and datasets. To date, our experiments have spanned a wide range of tasks, including image classification, node classification, link prediction, machine translation, and language modeling. The tested models range from simple architectures to complex ones, including MLPs, Graph Neural Networks (Hyperbolic), Transformers (Euclidean and Hyperbolic), the OPT model family (125M, 350M, 1B), and the LLaMA-1 model family (130M, 1.3B, 7B). Across all these scenarios, SST outperforms SOTA low-rank pre-training methods and achieves results comparable to, or even surpassing, full-rank counterparts, as demonstrated below.

To further assist you and minimize the effort required to review all the discussions, we have provided a brief summary of the key points raised and addressed during the review process:

1. **Strengths**:
    - The problem this paper is focusing on, i.e. memory-efficient LLM pre-training, is **important** and difficult.
      - Reviewer cEiN: "focused on an important problem"
    - Our proposed method, SST, is **novel**.
      - Reviewer 3vN4: "introduces an interesting approach"
      - Reviewer cEiN: “propose a novel memory-efficient LLM pre-training method”
      - Reviewer MMzE: "the design of SST is quite novel"
    - SST adopts a more effective approach by updating **all** singular values at each step. SST **selective** updates of singular vectors sampled from a multinomial distribution weighted by the magnitude of the singular values. Additionally, SST uses **singular value decomposition to initialize and reinitialize** low-rank parameters during training, reducing distortion relative to full-rank training compared to other low-rank methods.
      - Reviewer 3vN4: "updates all singular values of network weights and selectively updates singular vectors based on their significance"
      - Reviewer cEiN: "employs singular value decomposition to initialize and periodically reinitialize lowrank parameters"
      - Reviewer MMzE: "updates all singular values and selectively updates singular vectors"
      - Reviewer TBSo: "focusing on selective updates"
    - On various experiments, SST shows much better performance compared with other low-rank methods. SST reduces performance gap by **97.4%** (LLaMA-1.3B), **>100%** (LLaMA-7B), **65.8%** (LLaMA-125M, OpenWebText), **67.6%** (OPT-125M), **52.4%** (OPT-350M), **50.0%** (OPT-1.3B), **66.7%** (Transformer), **73.7%** (node classification), **82.5%** (link prediction), and **49%** (image classification). SST surpasses full-rank training with *1/8* trainable parameters (LLaMA-7B, until current steps), *1/4* (Transformer), and *1/8* (Hyperbolic Transformer).
      - Reviewer 3vN4: "often outperforms existing memory reduction ... occasionally matches or exceeds full-rank training ..."
      - Reviewer cEiN: "achieve a better performance than LoRA and its variants"
      - Reviewer MMzE: (1) "on natural language generation ... image classification, SST outperform existing memory reduction ... and is comparable to full-rank training in various cases"; (2) "show promising performance of SST"
      - Reviewer TBSo: (1) "outperforms current low-rank"; (2) "reduces the perplexity and BLEU score gaps"; (3) "often approaching the effect of full-rank training"
    - SST addresses and solves issues in prior low-rank methods.
      - Reviewer MMzE: (1) "reducing distortion"; (2) "clear discussion on the disadvantages of the baseline methods, such as ReLoRA"
      - Reviewer TBSo: "alleviate the saddle point problem of ReLoRA, bringing its training dynamics closer to full-rank training"
    - The presentation is clear and accessible.
      - Reviewer cEiN: "easy to follow"

2. **Questions**:
   - Question: Require for larger model (Reviewer 3vN4, cEiN)
    Reply: We trained full-rank, LoRA, ReLoRA and SST on LLaMA-1.3b (**Global Reply 1**) and LLaMA-7b (**Global Reply 4**). On LLaMA-1.3B, with 1/16 rank, SST achieves a PPL of **14.59** vs **14.54** for full-rank training. On LLaMA-7B, with 1/16 rank, until 40K steps, SST achieves a PPL of **19.66** vs **19.76** for full-rank training. These findings demonstrate SST’s scalability and effectiveness when applied to larger models.

---

> ### Author Response · Authors · 2024-12-04
> **Summary for Area Chair and Reviewers of Submission 6242 (2/3)**
>
> - Questions:
>   - Question: Why we need sampling updates (Reviewer cEiN)
>    Reply: We add an additional experiment (**Reply cEiN 1/4**) to compare sampling updates with always updating top-r singular vectors. Result shows that always updating top-r singular vectors causes the search space to degrade into a low-rank subspace. Sparse updates are crucial for pre-training, as they encourage the model to explore diverse directions.
>    - Question: Influence of sampling strategy (Reviewer 3vN4, TBSo)
>    Reply: Same as above, we add an additional experiment (**Reply 3vN4 2/2**) to compare MULTINOMIAL with other selecting mechanisms and show the importance of exploration.
>    - Question: Additional training cost from SST (Reviewer cEiN)
>    Reply: We truncate 20% training step of SST to achieve fewer training time than all the other methods. Result shows that SST still performs better than other method with full-training steps. (**Reply cEiN 2/4**)
>    - Question: Sensitivity analysis of the proposed method about hyper-parameters, such as the number of iterations and iteration interval (Reviewer cEiN, TBSo)
>    Reply: Some are already included in the original article. Others are conducted during rebuttal. (**Reply cEiN 2/4**)
>    - Question: Require for larger dataset (Reviewer cEiN)
>    Reply: We clarify that OpenWebText, the dataset we used in the article, is already very large, which could only be covered by one epoch for longest training. Additionally, we tested on C4, 25x larger than OpenWebText, and still demonstrate superior performance of SST. (**Reply cEiN 3/4**)
>    - Question: Hyper-parameter search for other baselines (Reviewer cEiN)
>    Reply: On LLaMA-130M, we tested LoRA and ReLoRA* with lr=1e-3 and lr=3e-3. We demonstrate SST outperforms them with both learning rate. (**Reply cEiN 3/4**)
>    - New Question: Performance gap between SST and full-rank training (Reviewer cEiN)
>    Reply: On LLaMA-1.3B, LLaMA-7B, and Transformer, even with **low rank**, SST is comparable with full-rank training. We believe that achieving comparable performance to full-rank training in low-rank settings is a more challenging and impactful goal than achieving performance parity in high-rank scenarios, which could benefit most resource-constrained researchers in the community. (**New Reply cEiN**)
>    - New Question: Performance gap narrows as rank increases (Reviewer cEiN)
>    Reply: The absolute performance gap decrease, but relative performance gap increase. (**New Reply cEiN**)
>    - Question: The overall framework and algorithm design are similar to ReLoRA (Reviewer MMzE)
>    Reply: Different from ReLoRA in even the core philosophy, where ReLoRA uses initialization-update-merging, but SST uses sampling-update-swapping. Sampling from existing singular vectors greatly benefits pre-training by avoiding the need to start from random or zero-initialized low-rank matrices. (**Reply MMzE 1/2**)
>    - Question: ReLoRA achieves quite comparable results with lower memory consumption  (Reviewer MMzE)
>    Reply: We jointly consider memory reduction and PPL increasement by defining a new metric called efficiency ratio, which represents how much percentage of memory can be reduced at the cost of a 1% PPL increase. SST improves the efficiency ratio by **167.4%** (OpenWebText, LLaMA-130M), **99.7%** (C4, LLaMA-130M), **196.1%** (OpenWebText, OPT-125M), **142.3%** (OpenWebText, OPT-350M), **65.9%** (OpenWebText, OPT-1.3B), and **4434.3%** (OpenWebText, LLaMA-1.3B). (**Reply MMzE 2/2**)
>    - Question: SVD affect scalablility (Reviewer TBSo)
>    Reply: The peak GPU memory consumption for SVD accounts for only **3%** of the total memory usage, and the time spent on SVD reinitialization is less than **1%** of the total training time. (**Reply TBSo 1/3**)
>    - Question:  As with other low-rank techniques, stability issues can still occur in very low-rank models even with the sampling mechanism of SST (Reviewer TBSo)
>    Reply: We are not sure what stability reviewer refer to. If instability refers to a sudden increase in training loss, SST already avoid it. If reviewer refers to stability as robustness, we conducted an additional experiment to show SST is more robust than other low-rank methods as rank decrease. (**Reply TBSo 1/3**)
>    - Question: Implementation of SST may be challenging (Reviewer TBSo)
>    Reply: We explain all the complexity is encapsulated within a class. This class seamlessly replaces the `nn.Linear` layer in the original model. And we plan to integrate SST into the PEFT library after acceptance. (**Reply TBSo 1/3**)
>    - Question: Is it possible that SST requires more training steps to achieve convergence compared to full-rank training? (Reviewer TBSo)
>    We conduct an additional experiment to show that even with rank equal to 1/16 of dimension, SST could catch up with full-rank training with more training steps. (**Reply TBSo 2/3**)
>
>  Best regards,
> *The authors of Submission 6242*

---

> ### Author Response · Authors · 2024-12-04
> **Summary for Area Chair and Reviewers of Submission 6242 (3/3)**
>
> Links:
> - LLaMA-1b eval loss curve: https://anonymous.4open.science/r/SST-iclr2025-review-2DB3/llama-1b.png
> - LLaMA-7b eval loss curve: https://anonymous.4open.science/r/SST-iclr2025-review-2DB3/llama-7b.png
> - Efficiency ratio: https://anonymous.4open.science/r/SST-iclr2025-review-2DB3/memory_vs_performance_ratio.pdf

---

### Meta-Review · Area_Chair_LxZh · 2024-12-10

**Metareview:**

This paper studies a memory-efficient training approach called Sparse Spectral Training (SST). The method selectively updates the singular vectors of weight matrices based on importance scores and demonstrates greater training stability compared to other low-rank update methods. Through experiments on Euclidean and Hyperbolic neural networks, the study shows that the proposed approach outperforms several other memory-reduced training approaches.

Strengths:
- Easy-to-follow methods.
- Promising performance compared to baselines.
- Studies of hyperbolic space.

Weaknesses:
- Limited novelty compared to baselines.
- Limited improvement compared to baselines.
- Pretraining results are mainly limited to small-scale high-perplexity/low average-score benchmarks.
- Missing reference and limited comparison to prior work, e.g., PISSA.
- Gap between SST and full-rank training.
- Much larger GPU memory consumption.

My main concern is that the authors need to explain why this is a better method for memory consumption and training efficiency in plain words in the method section and in experiments. I don’t quite understand, but why are these results buried in Appendix I and K, with very limited mention in the main paper? Table 22 says training time, but SST does not seem to provide much gain. There is also a limited comparison of these different methods on the BLEU-memory-time Pareto front, with Figure 6 being an exception (with only one baseline).

**Additional Comments On Reviewer Discussion:**

The authors clearly addressed several questions proposed by the reviewers. These include something like the following:
- More baselines on well-trained language modeling tasks. Scaling to larger datasets.
- Benchmarking different sampling approaches on singular vectors.
- Comparison to PISSA.
- Hyperparameter tuning on the number of iterations and iteration interval.

I suggest the authors significantly strengthen their discussions on the related papers in this area and highlight the novelty compared to them in future submissions.

---

### Decision · Program_Chairs · 2025-01-22

Reject